



# Characterizing model errors in chemical transport modeling of methane: Impact of model resolution in versions v9-02 of GEOS-Chem and v35j of its adjoint model

Ilya Stanevich[1], Dylan B. A. Jones[1], Kimberly Strong[1], Robert J. Parker[2,3], Hartmut Boesch[2,3], Debra Wunch[1], Justus Notholt[4], Christof Petri[4], Thorsten Warneke[4], Ralf Sussmann[5], Matthias Schneider[6], Frank Hase[6], Rigel Kivi[7], Nicholas M. Deutscher[8], Voltaire A. Velazco[8], Kaley A. Walker[1], and Feng Deng[1]

[1]Department of Physics, University of Toronto, Toronto, Canada
[2]Earth Observation Science, Department of Physics and Astronomy, University of Leicester, Leicester, UK
[3]National Centre for Earth Observation (NCEO) University of Leicester, Leicester, UK
[4]Institute of Environmental Physics, University of Bremen, Bremen, Germany
[5]Karlsruhe Institute of Technology (KIT), Institute of Meteorology and Climate Research (IMK-IFU), Garmisch-Partenkirchen, Germany
[6]Karlsruhe Institute of Technology (KIT), Institute of Meteorology and Climate Research (IMK-ASF), Karlsruhe, Germany
[7]Finnish Meteorological Institute, Sodankyla, Finland
[8]Centre for Atmospheric Chemistry, School of Chemistry, University of Wollongong, Wollongong, NSW, Australia

*Correspondence to:* Ilya Stanevich (stanevich@atmosp.physics.utoronto.ca)

**Abstract.**

The GEOS-Chem simulation of atmospheric $CH_4$ was evaluated against observations from the Thermal And Near infrared Sensor for carbon Observations Fourier Transform Spectrometer (TANSO-FTS) on the Greenhouse gases Observing SATellite (GOSAT), the Atmospheric Chemistry Experiment Fourier Transform Spectrometer (ACE-FTS), and the Total Carbon Column Observing Network (TCCON). We focused on the model simulations at the $4° \times 5°$ and $2° \times 2.5°$ horizontal resolutions for the period of February-May 2010. Compared to the GOSAT, TCCON, and ACE-FTS data, we found that the $2° \times 2.5°$ model produced a better simulation of $CH_4$, with smaller biases and a higher correlation to the independent data. We found large resolution-dependent differences such as a latitude-dependent $XCH_4$ bias, with higher columns abundances of $CH_4$ at high latitudes and lower abundances at low latitudes at the $4° \times 5°$ resolution than at $2° \times 2.5°$. We also found large differences in $CH_4$ column abundances between the two resolutions over major source regions such as China. These differences resulted in up to 30% differences in inferred regional $CH_4$ emission estimates from the two model resolutions. We performed several experiments using $^{222}$Rn, $^{7}$Be and $CH_4$ to determine the origins of the resolution-dependent errors. The results suggested that the major source of the latitude-dependent errors is excessive mixing in the upper troposphere and lower stratosphere, including mixing at the edge of the polar vortex, that is pronounced at the $4° \times 5°$ resolution. At the coarser resolution, there is weakened vertical transport in the troposphere at mid- to high latitudes due to the loss of sub-grid tracer eddy mass flux in the storm track regions. We also identified reduced vertical transport at the coarser resolution. The vertical air mass fluxes are calculated in the model from the degraded coarse-resolution wind fields and the model does not conserve the air mass flux




between model resolutions; as a result, the low resolution does not fully capture the vertical transport. This produces significant localized discrepancies, such as much greater $CH_4$ abundances in the lower troposphere over China at $4° \times 5°$ than at $2° \times 2.5°$. Although we found that the $CH_4$ simulation is significantly better at $2° \times 2.5°$ than at $4° \times 5°$, biases may still be present at $2° \times 2.5°$ resolution. Their importance, particularly, in regards to inverse modeling of $CH_4$ emissions, should be evaluated

in future studies using on-line transport in the native general circulation model as a benchmark simulation.

## 1 Introduction

Chemical transport models (CTMs) are useful tools to investigate the changing chemical composition of the atmosphere. They are driven by pre-calculated meteorological fields that may come from a free-running general circulation model (GCM), but are usually from re-analyses that assimilate atmospheric observations to provide a realistic description of atmospheric transport. It

is because of this ability to exploit reanalysis fields that CTMs are widely used for inverse modelling of surface emissions of environmentally important trace gases such as carbon monoxide (CO), carbon dioxide ($CO_2$), and methane ($CH_4$). But in this inverse modelling context, model errors become a major issue (Arellano and Hess, 2006; Baker et al., 2006; Chevallier et al., 2010; Houweling et al., 2010; Jiang et al., 2011; Patra et al., 2011; Locatelli et al., 2013; Chevallier et al., 2014; Houweling et al., 2015). For example, Locatelli et al. (2013) found that model errors could contribute to discrepancies that are as large

as 23%-48% in regional $CH_4$ source estimates, and as much as 150% in source estimates at the model grid-scale. However, characterizing and mitigating these errors is challenging.

In a companion analysis, Stanevich et al. (submitted) used a weak constraint (WC) four-dimensional variational (4D-Var) assimilation scheme to estimate errors in the $CH_4$ simulation in the GEOS-Chem CTM. They identified large biases in the high-latitude lower stratosphere, which the WC 4D-Var assimilation system was able to significantly reduce. The WC 4D-Var

scheme also identified an issue with vertical transport over the main continental source regions. For example, for eastern North America and East Asia, the estimated corrections to the $CH_4$ distribution exhibited a dipole structure with decreases in $CH_4$ over the source region and increases downwind, suggesting that too much $CH_4$ is confined close to the source region and not enough is exported from the continental source region. Stanevich et al. (submitted) found that the pattern of model corrections was similar, but weaker in magnitude when the model resolution was increased from $4° \times 5°$ to $2° \times 2.5°$, indicating that the

bias is influenced by the model resolution. Here we extend the analysis of Stanevich et al. (submitted) to examine specifically the impact of model resolution on the GEOS-Chem $CH_4$ simulation. Yu et al. (2017) recently investigated the impact of model resolution on transport in GEOS-Chem using $^{222}Rn$, $^{210}Pb$, and $^{7}Be$ tracers, and found that vertical transport is reduced in the model at coarse resolution. The analysis presented here complements Yu et al. (2017), with a specific focus on the impact of model resolution on the $CH_4$ simulation and the goal of better understanding the source of the biases identified in Stanevich

et al. (submitted).

To reduce computational costs, CTMs use meteorological fields that are archived at lower spatial and temporal resolution than the native resolution of the parent GCM that produced the fields. However, this creates a number of issues in the CTM. For example, this can lead to inconsistencies between archived horizontal winds (or air mass fluxes) and surface pressures (Jöckel





et al., 2001), which can result in the violation of mass conservation in the advection scheme. This lack of mass conservation is typically corrected using a mass (pressure) fixer (Bregman et al., 2003; Segers et al., 2002; Rotman et al., 2004). The degradation of the spatial and temporal resolution and the temporal averaging of the meteorological fields is also associated with some loss of information about eddy transport and, consequently, the weakening of vertical motion in the model (Grell

and Baklanov, 2011; Yu et al., 2017). Another consequence of reduced model resolution is increased numerical diffusion in the advection scheme, which can lead to rapid destruction of tracer gradients. Tracer advection in CTMs is often implemented using finite volume (FV) schemes because they are physically-based, conserve tracer mass by design and produce smooth solutions. However, diffusivity of an advection scheme depends on how well it approximates the sub-grid tracer distribution and how much of this information is transferred to the next time step. For example, as shown by Prather et al. (2008), the two

commonly used FV schemes, the second-order moments (SOM) algorithm (Prather, 1986) and the Lin-Rood (LR) scheme (Lin and Rood, 1996), have different effects on simulated tracer fields with coarsening of model resolution. The SOM scheme is less diffusive as it keeps two moments of the sub-grid tracer distribution, while the LR does not maintain any of the moments, but produces a new sub-grid tracer distribution each time based on the tracer concentrations in adjacent cells. Therefore, the SOM scheme is less sensitive to coarsening of model resolution and is better at preserving strong concentration gradients, such

as those at the edges of the polar vortex (Searle et al., 1998). Furthermore, Prather et al. (2008), compared the performance of the SOM and LR schemes and found that the SOM scheme showed monotonic convergence to the true solution with weak sensitivity to model resolution in the range from $4° \times 5°$ to $1° \times 1.25°$. The LR scheme did not show similar convergence, but the differences between the two schemes were significantly reduced as the resolution of the LR scheme was doubled from $4° \times 5°$ to $2° \times 2.5°$. Still, the discrepancies between the SOM and LR schemes at doubled resolution were larger than that

between the different resolutions of the SOM scheme, highlighting the diffusivity issues of the LR scheme at $4° \times 5°$.

Coarse vertical resolution can also be a significant issue in the tropopause region. Unresolved winds and increased mixing due to numerical diffusion can enhance stratosphere-troposphere exchange (STE) and significantly bias the vertical distribution of atmospheric gases (Locatelli et al., 2015). In the stratosphere, sharp tracer gradients are found at the borders of dynamic transport barriers such as the tropopause, polar vortex and stratospheric tropical pipe. Transport of chemicals in the presence

of the polar vortex has been investigated at different horizontal resolutions and using different advection schemes. Searle et al. (1998) assessed ozone loss in the Arctic polar vortex in the SLIMCAT CTM driven by the SOM advection scheme. They found weak sensitivity of ozone loss to model resolutions in the range from $1.4° \times 1.4°$ to $5.6° \times 5.6°$ resolution. Bregman et al. (2006) showed that the SOM scheme at $3° \times 2°$ resolution performed as well as a more diffusive first-order moments scheme at $1° \times 1°$ resolution. Strahan and Polansky (2006) evaluated the impact of different horizontal resolutions on the isolation of

the polar vortex in the Goddard Space Flight Center (GSFC) three-dimensional CTM, which uses the LR advection scheme. They focused on $CH_4$, ozone, and the age of air and found that $4° \times 5°$ resolution allowed too much mixing through the edges of the polar vortex. Significant improvement was achieved by doubling the resolution to $2° \times 2.5°$, with little sensitivity to further doubling ($1° \times 1.25°$), although the $1° \times 1.25°$ winds field were generated by linearly interpolating from the $2° \times 2.5°$ resolution. Strahan and Polansky (2006) also showed that at $4° \times 5°$, too much air leaked from the tropical pipe. Here we extend the analysis of Strahan and Polansky (2006) and focus on the impact of model resolution on the $CH_4$ simulation





in the troposphere and stratosphere in the GEOS-Chem CTM. There have been a number of inverse modelling studies trying to quantify global $CH_4$ emissions, and the majority of these have utilized models at coarse horizontal resolution ranging from about $2° \times 2°$ to $4° \times 6°$ (Chen and Prinn, 2006; Meirink et al., 2008; Bergamaschi et al., 2009, 2013; Fraser et al., 2013;

Cressot et al., 2014; Houweling et al., 2014; Monteil et al., 2013; Bruhwiler et al., 2014; Alexe et al., 2015; Locatelli et al., 2015; Feng et al., 2017). Because these different studies used different advection schemes, it is not possible to make a general statement about the quality of transport in the models. However, our goal here is to quantitatively characterize the resolution-induced errors in the GEOS-Chem $CH_4$ simulation and assess their potential implications for the use of GEOS-Chem for $CH_4$ source inversion analyses.

The paper is organized as follows. In Sect. 2, we describe the model and data sets used in the analysis. In Sect. 3 we evaluate the forward model simulation at $2° \times 2.5°$ and $4° \times 5°$ using different sets of observations. We then assess the impact of the resolution-induced model biases on optimized $CH_4$ surface emissions in Sect. 4. In Sect. 5, we investigate the origin of the model errors in the troposphere and the stratosphere using a set of transport tracers, and discuss possible mechanisms responsible for the biases. Finally, in Sect. 6, we present a summary and discussion of our results.

## 2 Model and Data

### 2.1 The GEOS-Chem model

In our analysis, forward model simulations of $CH_4$ are conducted using v9-02 of the forward GEOS-Chem model (www.geos-chem.org), while $CH_4$ emissions are estimated using v35j of the GEOS-Chem adjoint model (Henze et al., 2007), which is based on v8-02-01 of the forward model with updates from v9-02. GEOS-Chem is driven by archived meteorological fields

from the Goddard Earth Observing System (GEOS-5) produced by the NASA Global Modelling and Assimilation Office (GMAO) Data Assimilation System. We use version GEOS-5.2.0 of the meteorological fields, which have a native horizontal resolution of $0.5° \times 0.667°$ with 72 hybrid-sigma vertical levels with a model lid at $0.01$ hPa. Winds are archived as 6-hourly averages while surface pressures are given as instantaneous fields every 6 hours. Meteorological fields are degraded horizontally to $4° \times 5°$ or $2° \times 2.5°$ for the global simulations in GEOS-Chem. In the nested version of the model, the

simulation is performed at the native GEOS-5 horizontal resolution over selected domains, such as North America, Europe, and Asia. The vertical grid is reduced to 47 levels by combining layers above about 80 hPa. The final vertical spacing in the 47-level version of the model ranges from about $150$ m near the surface to about $1$ km in the upper troposphere and lower stratosphere (UTLS) and about $4$ km in the upper stratosphere. The meteorological fields are interpolated to the internal GEOS-Chem transport time step of 30 min for the $4° \times 5°$ grid and to 15 min for the $2° \times 2.5°$ resolution. In the nested version of

GEOS-Chem the transport time step is 10 min.

Advection is conducted using the Lin-Rood scheme (Lin and Rood, 1996). Tracer fields at cell edges in the horizontal directions are reconstructed using the the piece-wise parabolic method (Colella and Woodward, 1984) with a full monotonicity constraint (eliminates both overshoots and undershoots). A quasi-monotonic method with Huynh's $2^{nd}$ monotonicity constraint (Lin, 2004; Huynh, 1997) is applied in the vertical. Vertical advection at the coarse resolution is performed on a fixed hybrid-





sigma grid where the pressure thickness of the layers changes with surface pressure. In the nested domain at high resolution advective transport is limited to two dimensions and is performed between two floating vertical Lagrangian surfaces. The tracer distribution is then regridded in a mass conserving way from the predicted Lagrangian surfaces to a Eulerian hybrid-sigma

vertical grid at each transport time step. At high latitudes, at each vertical level, when the Courant number in the $x$-direction ($C^x$) becomes larger than unity the algorithm switches to a flux-form semi-Lagrangian scheme, which makes the advection scheme stable for longer time steps. For transport across polar caps the tracer fields are averaged over the two most northern (or southern) latitudinal bands.

Vertical air mass fluxes (AMFs) are calculated based on the divergence of horizontal air mass fluxes defined at the cell

edges. Horizontal AMFs at cell edges are calculated from the pressure and horizontal wind fields at the coarse resolution, which are degraded from the native resolution cell-centred winds using a surface pressure weighting procedure (Wang et al., 2004). The cell-averaged winds in the inner advective form operators in the Lin-Rood scheme are also derived from the airmass fluxes at the cell edges. Horizontal air mass fluxes are also corrected using a "pressure fixer" (Rotman et al., 2004) so that the vertical integral of air mass flux divergence remains consistent with the surface pressure tendency in each surface

cell. Convection is performed by a moist convective plume scheme that is driven by upward convective mass fluxes and mass detrainment rates (mass deposition rates into each layer) from the GEOS-5 relaxed Arakawa-Schubert scheme (Moorthi and Suarez, 1992). Coarse resolution fields are obtained by conservative regridding of the convective mass fluxes from the native GEOS-5 resolution. Mixing in the boundary layer takes place instantaneously from the surface to the height of the mixed layer, which is taken from the archived GEOS-5 fields.

Emissions of $CH_4$ at the surface are from a variety of biogenic and anthropogenic sources. A detailed description of $CH_4$ sources and sinks in GEOS-Chem can be found in Wecht et al. (2014). Anthropogenic sources include $CH_4$ emissions from natural gas and oil extraction, coal mining, livestock, landfills, waste water treatment, rice cultivation, biofuel burning and other minor sources, and are based on the 2004 anthropogenic inventory from the Emission Database for Global Atmospheric Research (EDGAR) v4.2 (European Commission Joint Research Centre/Netherlands Environmental Assessment Agency, 2009).

Natural $CH_4$ sources in GEOS-Chem include wetland emissions (Kaplan, 2002; Pickett-Heaps et al., 2011), termite emissions (Fung et al., 1991) and fire emissions from the Global Fire Emissions Database Version 3 (GFED3, van der Werf et al., 2010; Mu et al., 2011). Total $CH_4$ emissions at different model resolutions are slightly different due to the non-linear dependence of wetland emissions on the meteorological fields. In order to conserve the total emissions and to separate the effect of transport on the $CH_4$ fields, we regridded the coarse ($4° \times 5°$) resolution emissions to the finer grids. The majority of $CH_4$ loss is due to

oxidation by OH in the atmosphere. Tropospheric OH fields are prescribed as a three-dimensional monthly mean climatology, which results in a tropospheric $CH_4$ lifetime of 9.9 years (Wecht et al., 2014). Stratospheric $CH_4$ loss is estimated based on an archived climatology of $CH_4$ loss frequencies from the NASA Global Modeling Initiative (GMI) model (Murray et al., 2012). The remaining $CH_4$ sink is due to soil absorption. The total modelled $CH_4$ lifetime is 8.9 years (Wecht et al., 2014).





## 2.2 Datasets

### 2.2.1 GOSAT

In this study, we used shortwave-infrared $XCH_4$ retrievals from the Thermal And Near infrared Sensor for carbon Observations Fourier Transform Spectrometer (TANSO-FTS) on-board the Greenhouse gases Observing SATellite (GOSAT) (Parker et al., 2015). The satellite has a three-day repeat orbit period. TANSO-FTS has a surface footprint of 10.5 km in diameter and records spectra at about 13:00 local time. Column-averaged dry-air mole fractions of $CH_4$ ($XCH_4$) are retrieved from TANSO-FTS of shortwave infrared radiation (SWIR). These column retrievals have limited sensitivity to the vertical distribution of $CH_4$. We used version 5.2 of the University of Leicester (UoL) GOSAT Proxy $XCH_4$ retrievals over land (Parker et al., 2011, 2015). In the Proxy method, simplified retrievals of $XCO_2$ and $XCH_4$ are obtained in spectral bands centred at 1.65 μm and 1.61 μm, respectively. Their ratio ($XCH_4/XCO_2$) is multiplied by modeled $XCO_2$ fields to obtain the final total column-averaged dry-air mole fraction of $CH_4$. The Proxy method provides significantly greater observational coverage, especially in tropical areas, compared to traditional retrievals, however it does not account for potential biases in the $CO_2$ fields. In this study, the modeled $CO_2$ fields were taken from a GEOS-Chem ($4° \times 5°$ resolution) inversion analysis to estimate $CO_2$ surface flux using GOSAT $CO_2$ retrievals over land (Deng et al., 2014). $XCH_4$ retrievals over Greenland and to the north of $75°N$ were excluded. GOSAT Proxy $XCH_4$ retrievals with the original modeled $XCO_2$ data were evaluated against co-located TCCON ground-based measurements and were found to contain random errors of 12.55 ppb and systematic errors of 4.8 ppb (Hewson et al., 2015).

### 2.2.2 ACE-FTS

The stratospheric $CH_4$ fields in GEOS-Chem were evaluated using the version 3.6 solar occultation $CH_4$ retrievals (Boone et al., 2013; Waymark et al., 2014) from the Atmospheric Chemistry Experiment Fourier Transform Spectrometer (ACE-FTS, Bernath et al., 2005) on-board SCISAT. The ACE-FTS instrument makes 15 occultations per day (for both sunrise and sunset) separated by about $24°$ longitude, covering an altitude range from the cloud tops in the upper troposphere up to about 150 km. ACE-FTS has a low horizontal resolution of about 300 km in the limb direction, and a vertical resolution of about 3 km at a tangent point 3000 km away from the satellite, determined by the instrument field-of-view. However, because of oversampling, the instrument has an effective vertical resolution of about 1 km (Bernath et al., 2005). We use version 3.6 of the ACE-FTS $CH_4$ retrievals. Version 3.6 only differs from version 3.5 in that a local computer was used to process v3.5 while a shared supercomputing system was used for v3.6. Version 3.5 was compared by Olsen et al. (2017) to MIPAS $CH_4$ vertical profiles when both satellites were coincident with TANSO-FTS. The study found small differences except in the tropics. The mean differences were larger than 20% below about 450 hPa and within 5% between 450 and 40 hPa.

### 2.2.3 TCCON

We also evaluated the model against $XCH_4$ retrievals from the Total Carbon Column Observing Network (TCCON) (Wunch et al., 2011). TCCON consists of a network of ground-based high-resolution Fourier Transform InfraRed (FTIR) spectrometers





retrieving XCH$_4$ from solar absorption spectra in the near-infrared region. We used GGG2014 version of the data from multiple TCCON stations (Kivi and Heikkinen, 2016; Kivi et al., 2017; Blumenstock et al., 2017; Griffith et al., 2017; Hase et al., 2017; Notholt et al., 2017; Sherlock et al., 2017; Sussmann and Rettinger., 2017; Warneke et al., 2017; Wennberg et al., 2017b, a).

These data are tied to the WMO CH$_4$ scale through comparisons with calibrated aircraft and Aircore profile measurements. The estimated accuracy and precision of XCH$_4$ retrievals are better than 0.5% and 0.3%, respectively (Wunch et al., 2015).

## 3    GEOS-Chem model validation

### 3.1    Comparisons with GOSAT

We simulated CH$_4$ using GEOS-Chem for the period from 1 February 2010 to 31 May 2010 at the two horizontal resolutions

of $4° \times 5°$ and $2° \times 2.5°$. The time period was chosen to match the analysis period of Stanevich et al. (submitted). Also, for simulations over longer periods, incorrect a priori emissions can become the dominant source of model errors and confound our analysis to assess transport errors in the model. As in Stanevich et al. (submitted), CH$_4$ fields at both resolutions were spun up for about 5.5 years until July 2009. From July 2009 to January 2010, monthly mean surface emissions optimization was performed using the $4° \times 5°$ resolution model constrained by GOSAT Proxy XCH$_4$ retrievals. The regridded optimized

emissions were also used to perform the $2° \times 2.5°$ resolution simulation for the same period. The updated model fields on 1 February 2010 at both model resolutions were taken as initial conditions for the analysis period. All results were later converted to, and evaluated at, the $4° \times 5°$ resolution. Figure 1 shows the GOSAT XCH$_4$ data (first column) and the optimized XCO$_2$ fields (second column). The modelled CH$_4$ fields were smoothed with the GOSAT scene-dependent averaging kernels. The third and the fourth columns in the figure show the monthly mean differences between the simulated XCH$_4$ fields at the two

resolutions and the GOSAT retrievals. Both difference fields represent the combined effects of errors in transport, chemistry, and the emissions, as well as possible biases in the XCH$_4$ retrievals. The fact that these differences are smaller at the $2° \times 2.5°$ resolution than at $4° \times 5°$ implies that there are resolution-dependent transport errors in GEOS-Chem. This is consistent with the model corrections (the forcing terms) calculated by Stanevich et al. (submitted) using the GEOS-Chem weak constraint 4D-Var assimilation scheme. The difference between the CH$_4$ fields simulated at the two resolutions, co-located with GOSAT

observations and smoothed with GOSAT averaging kernels, is presented in the last column in Fig. 1. In general, it shows that at finer resolution, XCH$_4$ columns are smaller in the middle and high latitudes and larger in the tropics. There are also several regional features, such as a large positive XCH$_4$ bias over Northern Europe and Russia that has been reduced but not completely removed at the $2° \times 2.5°$ resolution. In addition, the XCH$_4$ fields simulated at the $2° \times 2.5°$ resolution have a smaller positive bias relative to the GOSAT observations over China.

One cannot say conclusively from this comparison if spatially dependent biases are still present in the $2° \times 2.5°$ simulation. The positive high-latitude bias relative to GOSAT in the northern hemisphere (NH) was smaller compared to the $4° \times 5°$ simulation. However, a weak positive bias over the polar vortex is still present (see Fig. 1). It is unclear whether this bias is related to transport errors or to possible systematic errors in the GOSAT retrievals under the polar vortex conditions (see Fig. 9). The possibility that a weak positive latitudinal bias is still present in the southern hemisphere (SH) in GEOS-Chem





at $2° \times 2.5°$ can be observed over southern South America, Australia, and the southern tip of Africa. Negative $XCH_4$ biases over the Himalayas and Andes were not affected by doubling the model resolution. They could be related to discrepancies in surface pressure between GEOS-Chem and the GOSAT retrievals, however, even vigorous filtering of GOSAT retrievals based on differences in surface pressure did not eliminate them. These biases may also indicate errors in $CH_4$ uplift over the

mountains. Other $XCH_4$ biases not significantly affected by model resolution are located over Africa, including a positive bias over western equatorial Africa and a negative bias over southeastern Africa. These biases, as well as biases over mountains, may also be related to the $XCO_2$ fields used in the GOSAT $CH_4$ retrievals. The $CO_2$ fields were obtained by assimilating GOSAT $XCO_2$ retrievals to constrain $CO_2$ fluxes in the coarse $4° \times 5°$ resolution GEOS-Chem model. If the $CO_2$ flux inversion did not remove all biases in the $CO_2$ state, the latter could be projected onto the $XCH_4$ retrievals. For example, the negative model

bias in $XCH_4$ in south-eastern Africa could be related to a positive bias in the $XCO_2$ data due to an overestimate of the regional $CO_2$ fluxes, which was then transferred to the $XCH_4$ retrievals.

### 3.2 Comparisons with ACE-FTS

Stanevich et al. (submitted) found large corrections to the modeled $CH_4$ fields in the lower stratosphere in the weak constraint 4D-Var assimilation, suggesting a possible stratospheric origin of the model errors. Zonal median differences between GEOS-

Chem and ACE-FTS in the stratosphere over the four-month period (February-May 2010) are shown in Fig. 2. The figure also shows a comparison of the modelled stratospheric $CH_4$ fields to the GOSAT a priori stratospheric $CH_4$ profiles, which come from the TOMCAT (Chipperfield, 2006) model with assimilated ACE-FTS retrievals (Parker and the GHG-CCI group, 2016). The figure indicates that the GEOS-Chem stratosphere at $4° \times 5°$ resolution is positively biased against ACE-FTS at middle to high latitudes and is weakly negatively biased in the tropics. A similar stratospheric bias was reported by **??**. The strongly

positive feature in the NH above $100 \, hPa$, poleward of $45°N$, is most likely related to the polar vortex. The mismatch relative to ACE-FTS was significantly reduced at the $2° \times 2.5°$ resolution. Furthermore, the $CH_4$ differences in the NH stratosphere at $2° \times 2.5°$ became even weakly negative, although a positive anomaly remained in the lower stratosphere and in the SH.

Figure 3 shows the impact of the modeled $CH_4$ mismatch relative the ACE-FTS above the tropopause on the $XCH_4$ fields. The $XCH_4$ difference between GEOS-Chem and ACE-FTS was obtained by augmenting the ACE-FTS profile in the strato-

sphere by the GEOS-Chem profiles in the troposphere and smoothing the resulting vertical $CH_4$ difference profile with zonal mean GOSAT averaging kernels, averaged in $4°$ latitudinal bands. This was done to compare the result to the biases observed in Fig. 1. We used only ACE-FTS profiles above the tropopause for each GEOS-Chem lon-lat grid cell for a particular time instance. The dynamic tropopause pressure was taken from the archived GEOS-5 meteorological fields. As in Fig. 2, the median zonal mean value of the $XCH_4$ differences was used to avoid the influence of possible outliers and the sparsity of ACE-FTS

retrievals. Fig. 3 indicates that the $XCH_4$ differences have a latitudinal structure. The differences are small in the tropics at both resolutions. In NH mid-latitudes ($30°-60°N$), they reach about $15 \, ppb$ at $4° \times 5°$, but are reduced to less than $5 \, ppb$ at the $2° \times 2.5°$ resolution. The SH in February-May appears to be less sensitive to model resolution and the reduced $XCH_4$ difference at $2° \times 2.5°$ resolution was 5 to 10 ppb in this region. We note that these results may be affected by the sparsity of the ACE-FTS measurements and possible biases in ACE-FTS $CH_4$ retrievals in the lower stratosphere. De Mazière et al.



(2008) suggested potential biases of 10% in the older version 2.2 of ACE-FTS $CH_4$ retrievals in the UTLS region. There is also a sharp positively biased feature in the SH stratosphere, the origin of which is unclear.

Generally, stratospheric $CH_4$ is not well mixed, and transport errors acting on $CH_4$ gradients may significantly affect the $CH_4$ fields. A number of studies have found discrepancies in stratospheric $CH_4$ fields between different CTMs, with mainly
positive biases between the models and observations (Patra et al., 2011; Ostler et al., 2016; Saad et al., 2016). Discrepancies are usually attributed to biases in the STE and the mean residual (Brewer-Dobson) circulation in the stratosphere, however, they may also be caused by a biased stratospheric $CH_4$ sink. Still, at coarse resolution, depending on the diffusivity of the numerical advection scheme, numerical diffusion, and hence increased STE, could be a major factor contributing to biases in the stratospheric $CH_4$ fields.

## 3.3   Comparisons with TCCON

The third component of the model evaulation involved comparisons to ground-based $XCH_4$ retrievals from TCCON (Wunch et al., 2011). Table 1 gives a summary of the evaluation of the two model resolutions against TCCON data. The results show that running the model at $2° \times 2.5°$ resolution significantly improved the correlations and reduced the positive model misfits against TCCON, except for the northern- and southern-most stations (Lauder, Wollongong and Sodankyla). For the $2° \times 2.5°$
resolution, there is a 7-8 ppb difference relative to Wollongong and Lauder in the SH and about a 15 ppb difference relative to Sodankyla in the NH. Similar differences were also observed in model$-$GOSAT comparisons (Fig. 1), with a 3-4 ppb difference at Wollongong and Lauder and about a 5 ppb difference at Sodankyla. Significant misfit reduction was achieved for Sodankyla (from 30 ppb to 15.6 ppb), and it is unclear whether the remaining difference is due to the model or the observations. As shown by Tukiainen et al. (2016) and Ostler et al. (2014), high latitude $XCH_4$ retrievals can be subject to systematic errors
during polar vortex conditions due to variability in $CH_4$ a priori profiles. As with the GOSAT and ACE-FTS comparisons, the TCCON results also suggest that a weak latitudinal bias may still be present in the model at the $2° \times 2.5°$ resolution and, given the results of the ACE-FTS comparisons and other work (Saad et al., 2016), it may be related to the stratosphere.

## 4   Impact on surface emissions

The primary reason we are concerned about the magnitude of biases in the model $XCH_4$ fields is their possible impact on
estimates of $CH_4$ surface emissions. As can be seen in Fig. 1 (fifth column), the bias between the modeled $CH_4$ at $4° \times 5°$ and at $2° \times 2.5°$ is comparable in magnitude to the mismatch between the $2° \times 2.5°$ $XCH_4$ fields and the GOSAT data. This suggests that model errors at $4° \times 5°$ resolution may have an adverse impact on optimized emissions. We therefore optimized monthly $CH_4$ surface emissions for the period of February-May 2010 using GEOS-Chem at the two resolutions. Surface emissions were optimized as monthly totals in each model grid box. Monthly emissions in all four months were adjusted simultaneously in order to best match the GOSAT $XCH_4$ measurements during the same period. We used the strong constraint 4D-Var approach, which was used by Stanevich et al. (submitted) and Wecht et al. (2014). The strong constraint 4D-Var data assimilation assumes that the model is perfect, except for potential biases in the $CH_4$ surface emissions, so that the forward





model equation can be written as

$$\mathbf{x}_{i+1} = M(\mathbf{x}_i, \mathbf{p}) \tag{1}$$

where the forward model operator $M$ acts on the model state $\mathbf{x}_i$ at the current time step $i$, with surface emissions $\mathbf{p}$, to produce a new model state $\mathbf{x}_{i+1}$ at the next time step $i+1$. The 4D-Var cost function is then defined as

$$
\begin{aligned}
J(\mathbf{p}) = &\sum_{i=0}^{N} \frac{1}{2}(\mathbf{y}_i - \mathbf{H}\mathbf{x}_i)^T \mathbf{R}_i^{-1}(\mathbf{y}_i - \mathbf{H}\mathbf{x}_i)+ \\
&+\frac{1}{2}(\mathbf{p} - \mathbf{p}_a)^T \mathbf{B}^{-1}(\mathbf{p} - \mathbf{p}_a)
\end{aligned}
\tag{2}
$$

with a constraint given by Eq. 1, where $\mathbf{y}_i$ are the CH$_4$ observations, $\mathbf{H}$ is the linear observation operator that maps the modelled CH$_4$ state into the measurement (XCH$_4$) space, and $\mathbf{R}_i$ represents the observational error covariance matrix. A priori estimates of the model parameters and their error covariance matrix are given by vector $\mathbf{p}_a$ and matrix $\mathbf{B}$, respectively. Both $\mathbf{R}$ and $\mathbf{B}$ were assumed to be diagonal. Uncertainty on the a priori emissions in each $4° \times 5°$ and $2° \times 2.5°$ grid box was set to 50% and 100%, respectively, in order to be approximately in accord with the central limit theorem. In designing $\mathbf{R}$, we used XCH$_4$ retrieval errors as the observational uncertainty and inflated them to match the global mean GOSAT standard deviation (scatter) against TCCON observations (Parker et al., 2015). The minimization of the cost function was performed using the L-BFGS-B algorithm (Byrd et al., 1995) and the adjoint of the GEOS-Chem model.

Figure 4 shows the ratio of optimized to a priori CH$_4$ emissions, which are referred to as emission scaling factors. The $4° \times 5°$ inversion suggested lower CH$_4$ emissions at high latitudes and higher emissions in the tropics compared to the $2° \times 2.5°$ inversion. The differences are particularly large over equatorial Africa and Europe. There are large reductions in the emissions across mid-latitude Eurasia at $4° \times 5°$, whereas the changes in this region were minor at $2° \times 2.5°$. Furthermore, at $2° \times 2.5°$ there were increases in the emissions in Europe.

The CH$_4$ emissions aggregated into the widely used 11 TransCom land regions (Gurney et al., 2004) are plotted in Fig. 5. Because of the particular division of the TransCom regions, the aggregated emissions at the two model resolutions were more similar to each other than suggested in Fig. 4. The largest changes are observed over Tropical South America where the $2° \times 2.5°$ inversion increased the a priori emissions by 30%, whereas the $4° \times 5°$ inversions increased them by 60%. Over Temperate South America, Europe and Boreal Eurasia, the $2° \times 2.5°$ emissions remained at their a priori level, whereas the $4° \times 5°$ emissions were reduced by 17%, 26% and 29%, respectively. Additionally, the resulting emissions over Northern Africa, which is partly comprised of wetland emissions from Equatorial Africa, were about 16% smaller at the $2° \times 2.5°$ resolution. It is likely that boreal North American emissions would be more biased, however the analysis covered only the time period when local wetland emissions were not significant. The moderate sensitivity of the emissions to the induced biases in the large TransCom regions could be due to the sparse observational coverage of GOSAT. With a greater observational constraint, such as from the TROPOMI (Veefkind et al., 2012) or the future MERLIN (Kiemle et al., 2014) satellite, the optimized emissions may be more sensitive to model errors.

Several studies have tried to address latitudinal biases in CH$_4$ in models or observations. For example, Turner et al. (2015) applied a mean uniform latitudinal correction to their modelled CH$_4$ fields prior to performing their inversion analysis, while





other studies (Fraser et al., 2013; Alexe et al., 2015) tried to fit a latitudinal correction in their inversion. While this may partly mitigate the problem, a latitudinal correction does not work well if the latitudinal bias is associated with biased initial

conditions and surface emissions, for example, due to systematic underestimation/overestimation of a priori $CH_4$ emissions in the tropics versus mid-latitudes. Moreover, as will be shown below, the actual latitudinal bias may vary in time and may not be zonally uniform, such as with features associated with the polar vortex.

## 5   Origin of Model Errors

### 5.1   Mean meridional circulation

To investigate the mechanisms responsible for the differences in transport between the $4° \times 5°$ and $2° \times 2.5°$ model resolutions, we used radon-222 ($^{222}$Rn) and beryllium-7 ($^{7}$Be), together with $CH_4$, as tracers of atmospheric transport. In these experiments, the sources and sinks of the tracers are identical across all model resolutions, so differences in the tracer fields are due solely to transport. $^{222}$Rn is emitted from soils by decay of radium-226 ($^{226}$Ra) and is lost in the atmosphere through radioactive decay to lead-210 ($^{210}$Pb), with a half-life of 3.8 days (Jacob et al., 1997). Due to its short lifetime and sources at the surface,

$^{222}$Rn is useful tracer of vertical transport in the troposphere. In GEOS-Chem, land and ocean fluxes of $^{222}$Rn are set to 1 and 0.005 $\mathrm{atoms\,cm^{-2}\,s^{-1}}$, respectively. Emissions over land in the polar regions beyond $60°$ are set to 0.005 $\mathrm{atoms\,cm^{-2}\,s^{-1}}$. Corresponding emissions are reduced threefold when surface temperatures are below zero. $^{7}$Be is produced in the atmosphere (mainly in the stratosphere) in a process of spallation of nitrogen and oxygen atoms by cosmic ray bombardment (Lal and Peters, 1967) and is immediately attached to aerosol particles. It is removed from the atmosphere by radioactive decay with a

half-life of 53.3 days and by dry deposition and wet scavenging (Liu et al., 2001). $^{7}$Be sources in GEOS-Chem are prescribed following Lal and Peters (1967) and Liu et al. (2001). We used $^{7}$Be (as well as $CH_4$) as a tracer of stratosphere-troposphere exchange (Liu et al., 2016) and stratospheric mixing. However, it can also be useful indicator of tropospheric descent.

    The experiments were run for two months (February-March 2010), with the first month used as a spin-up period. The first row in Fig. 6 gives the mean monthly zonal tracer distribution modeled at $4° \times 5°$ resolution while the second row presents

the zonal mean differences between the $2° \times 2.5°$ and $4° \times 5°$ simulations. The $^{222}$Rn experiment indicates that at $4° \times 5°$ there is up to a 40% reduction in the tracer concentrations in the middle and upper troposphere relative to $2° \times 2.5°$, with a noticeable increase in the tracer concentrations in the lower troposphere ranging from 10% to 25% (see Fig. 6d). These results suggest that at coarser resolution, vertical transport in GEOS-Chem is reduced. These results are similar to those shown by Yu et al. (2017). Discrepancies in the tracer concentrations are aligned along isentropes and are, therefore, most likely related

to adiabatic transport as part of the large-scale circulation associated with extratropical cyclones. The tracers are lifted in warm conveyor belts that are linked with these cyclones (Stohl, 2001; Eckhardt et al., 2004; Hess, 2005; Parazoo et al., 2011). However, discrepancies in orographic and convergent uplift at mid-latitudes may also contribute to the total bias. Reduced vertical transport causes $^{222}$Rn to remain near the surface and in the cold pocket at lower potential temperature surfaces. The largest fractional overestimate of $^{222}$Rn at $4° \times 5°$ resolution is aligned with the 265 K potential temperature surface, wheras

the largest fractional underestimate corresponds to the 310 K surface. There are also large fractional differences in the





stratosphere, although the $^{222}$Rn concentrations are low there. At $2° \times 2.5°$, there is much less $^{222}$Rn in the high-latitude lower stratosphere and more in the tropics and in the upper stratosphere. These differences may point to increased cross-tropopause transport at coarser resolution, which may also contribute to lower $^{222}$Rn concentrations in the upper troposphere (UT) at $4° \times 5°$. They may also indicate increased isentropic mixing through the boundaries of the tropical pipe in the stratosphere,

transporting more air with high levels of $^{222}$Rn out of the tropical pipe in the $4° \times 5°$ simulation.

$^{7}$Be in the $4° \times 5°$ resolution simulation was reduced by up to 20% relative to the $2° \times 2.5°$ simulation in the extratropical lower stratosphere (LS) and increased by up to 20% in the tropical UT (see Fig. 6e). The extra-tropical LS interacts with the UT across the tropopause through isentropic mixing. Air in the UTLS region is stirred and subsequently mixed by cutoff cyclones, tropopause folds and uplift in warm conveyor belts of extratropical cyclones (Stohl et al., 2003). The $^{7}$Be experiment indicates

that STE was enhanced at $4° \times 5°$ resolution. The results also suggest that subsidence in the troposphere between the 270 K and 300 K isentropes is stronger at the $2° \times 2.5°$ resolution, resulting in higher $^{7}$Be concentrations in the lower troposphere at mid-latitudes (up to 35%) and lower $^{7}$Be concentrations in the UT in polar regions (up to 20%).

The results of the CH$_4$ experiment were similar to those from the other two experiments. CH$_4$ is well mixed in the troposphere and, therefore, is not a good tracer of tropospheric transport. However, the difference plot in Fig. 6f shows similar

evidence of reduced vertical transport, with lower CH$_4$ concentrations below 900 hPa in the NH and higher concentrations in the free troposphere in the $2° \times 2.5°$ simulation than in the $4° \times 5°$ run. Higher CH$_4$ abundances in the middle and upper extratropical troposphere in the NH could also be explained by a stronger tropopause barrier at the $2° \times 2.5°$ resolution, which prevents CH$_4$ from mixing into the LS, leading to more CH$_4$ accumulation in the troposphere. Transport in the NH was more strongly affected by model resolution as CH$_4$ gradients are larger in this region. It could also be explained by the fact that the

results are given for the month of March when baroclinic wave activity is stronger in the NH. The ACE-FTS comparisons in Figs. 2 and 3 showed similar results, with the model biases in the NH more sensitive to changes in model resolution than in the SH.

## 5.2    Polar vortex

The zonal mean plots in Figs. 6d-f obscure some details of stratospheric transport related to the polar vortex dynamics. To

better understand the source of the stratospheric bias, we carried out yearly comparisons of simulated CH$_4$ fields at both model resolutions with a focus on the polar lower stratosphere. Simulations were run from 1 July 2009 to 1 July 2010 with the identical model setup and with the same initial conditions. Figures 7 and 8 show mean monthly CH$_4$ differences between the two resolutions at the 50 hPa pressure surface for the NH and SH, respectively. The dark blue colors correspond to the regions with lower CH$_4$ in the $2° \times 2.5°$ simulation. These regions of lower CH$_4$ are aligned with and evolve together with the polar

vortex, which is illustrated in the plot of potential vorticity in the NH that is shown in Fig. 9. A vivid example is the vortex splitting event in February 2010. As the polar vortex becomes stronger and develops sharper potential vorticity (PV) gradients at its boundaries, the latter become barriers for mixing. Subsidence inside the polar vortex brings air depleted in CH$_4$ down from higher altitudes. This air mixes inside the polar vortex but does not mix with the vortex exterior. However, as can be seen in Figs. 7-8, this barrier is weaker at $4° \times 5°$ and results in more CH$_4$ relative to $2° \times 2.5°$ resolution. As expected, the model





CH$_4$ differences are also aligned with the model−GOSAT XCH$_4$ differences shown in Fig. 1. In the case of the vortex splitting event in February 2010, these differences were observed over both North America and Eurasia.

### 5.3 XCH$_4$ bias

As described in Sect. 2, the number of vertical layers in the GEOS-Chem model, compared to original the GEOS5 model, can be reduced in the upper stratosphere above 80 hPa. There are 36 vertical levels in this region in the original metfields which are reduced to just 11 levels here, so that the total number of vertical levels is reduced from 72 to 47. We also assessed the impact of this vertical regridding on modelled XCH$_4$ fields. We ran the model with 72 and 47 vertical levels, at both horizontal resolutions. However, for the comparison, all simulations were regridded to the $4° × 5°$ resolution with 47 vertical levels. The

CH$_4$ differences between the model simulations were vertically smoothed with mean zonal GOSAT averaging kernels, which were averaged in $4°$ latitudinal bands over the entire period of the observations.

The purpose of these experiments was both to determine regional biases induced in the XCH$_4$ fields by transport errors and to assess their impact on the total atmospheric CH$_4$ budget. It was shown in previous sections that at $2° × 2.5°$, there is less CH$_4$ in the stratosphere and more in the troposphere. Such redistribution could result in additional CH$_4$ chemical loss as most

of the OH mass is located in the tropical troposphere. Hence, transport errors could potentially project onto an atmospheric CH$_4$ sink. However, the results did not show any noticeable sensitivity of the CH$_4$ budget to horizontal resolution and vertical resolution in the upper stratosphere. Changes in the total CH$_4$ burden in both cases were negligible and didn't exceed 0.05% over the 6-year period.

The differences in the CH$_4$ distributions between the experiments are presented in Fig. 10. They suggest that increased

model vertical resolution in the upper stratosphere had a modest impact on the CH$_4$ fields, which is consistent with Strahan and Polansky (2006). However, it should be noted that changes in the vertical resolution in the lower stratosphere may have a larger impact on the CH$_4$ distribution. For example, Locatelli et al. (2015) showed that increased vertical spacing in their model caused an additional leak of tropospheric CH$_4$ into the stratosphere, significantly biasing the vertical CH$_4$ distribution.

The results of the second experiment with doubling of the horizontal resolution from $4° × 5°$ to $2° × 2.5°$ showed that,

generally, at $2° × 2.5°$ CH$_4$ is reduced in the column at high latitudes to the north of about $40°$N and south of about $50°$S and is increased in tropics. As suggested by Fig. 6f, the general bias structure is mainly a result of CH$_4$ redistribution at $2° × 2.5°$, with more CH$_4$ in the troposphere and less CH$_4$ at high latitudes in the stratosphere. The XCH$_4$ increase in the tropics is largely in the SH, initially, and after four years the pattern of biases becomes more symmetric across the equator. The general bias pattern is also modulated by seasonally changing XCH$_4$ biases associated with the polar vortex in both hemispheres. In the

SH, where the polar vortex is stronger and more isolated, negative XCH$_4$ biases are sharp, localized and as large as -31 ppb, while in the NH, the bias fields are more diffused with maximum amplitude of -23 ppb. Strong seasonally varying negative XCH$_4$ biases in the polar regions are also accompanied by additional positive XCH$_4$ anomalies of up to 12 ppb in the tropics.





## 5.4 Vertical transport in the troposphere

The tracer experiments in Sect. 5.1, as well as the work of Yu et al. (2017), revealed issues in modelled vertical transport in the troposphere. Here, we investigate possible causes of the reduced vertical transport at the coarse model resolution. The structure of the $^{222}$Rn bias between the two model resolutions suggested errors in tracer advection at mid-latitudes. Here we examine possible sources of these transport errors and their impact on the modelled tracer fields.

### 5.4.1 Regridding mass flux versus horizontal winds

Figure 11 shows the mean spatial distribution of the vertical AMFs (vAMFs) at the 590 hPa pressure level in GEOS-Chem at $4°$ $\times 5°$ and at the native $0.5° \times 0.667°$ horizontal resolution in May 2010 over east Asia. They show that the high-resolution fluxes have more detailed structure due to the complex topography of China, and points to the potential of producing inconsistencies in calculating the vAMFs at different model resolutions due to the fact that, at coarse resolution, vAMFs are obtained from
degraded coarse resolution cell-centred winds (see Sect. 2.1). To quantify the potential impact of calculating the horizontal AMFs (hAMFs) at the grid cell interfaces from the coarse resolution wind fields, we consider using archived hAMFs from the native resolution GEOS-5.2.0 fields and regridding them to the lower resolution. This would ensure consistency between the hAMFs at different model resolutions and, therefore, preserve the derived vertical air mass fluxes (vAMFs). Unfortunately, neither the native resolution hAMFs nor the global surface pressure and winds fields at the native GEOS-5.2.0 resolution were
available for us, as GMAO has transitioned to version GEOS-5.7.2 of their assimilation system.

Since we do not have the global high-resolution fields, to assess the impact of using lower resolution winds to derived the vAMFs, we performed two experiments using the $2° \times 2.5°$ and nested GEOS-5.2.0 model fields. In one experiment (R1 experiment), the $2° \times 2.5°$ hAMFs at the grid cell boundaries were regridded to $4° \times 5°$. In the other experiment (R2 experiment), we used the native $0.5° \times 0.67°$ hAMFs from the nested model domains for North America, Europe, and Asia.
The regridded native resolution hAMFs in the R2 experiment were merged with the ones calculated at the coarse resolution over the rest of the globe. The hAMFs at $2° \times 2.5°$ and $0.5° \times 0.67°$ were calculated in GEOS-Chem, corrected using the pressure fixer, regridded and saved for each $4° \times 5°$ resolution model transport time step. This also allowed us to turn off the pressure fixer in the R1 experiment as the use of higher resolution hAMFs guaranteed mass conservation. The regridded and merged hAMFs were then used to drive advection at the coarse $4° \times 5°$ model resolution. Figures 6g-i and Figs. 6j-l
summarize the results of the R1 and R2 experiments, respectively. If all of the errors were due to the use of low resolution winds to calculate the vAMFs, then the "fixed" $4° \times 5°$ simulation should look like the $2° \times 2.5°$ simulation, and consequently the differences between the fixed $4° \times 5°$ and standard $4° \times 5°$ simulations should resemble differences between the $4° \times 5°$ and $2° \times 2.5°$ simulations. Indeed, the R1 results show that the initial differences between tracer fields at $4° \times 5°$ and $2° \times 2.5°$ resolutions (Figs. 6d-f) were partly due to inconsistent AMFs. Using the corrected hAMFs, we were able to partly
mitigate vertical transport errors. For example, $^{222}$Rn was increased in the upper troposphere by up to 12% in the "fixed" $4° \times 5°$ simulation (Fig. 6g). Still, the proposed model fix does not explain all the differences between the $4° \times 5°$ and $2° \times 2.5°$ model resolutions. The R2 results suggest that the induced AMF bias between $4° \times 5°$ and native resolution is even larger:





regionally regridded native resolution hAMFs had a strong impact on the $CH_4$ fields at the $4° \times 5°$ resolution in both the lower troposphere and the UTLS. The general feature of the results obtained is that the tropospheric correction to vertical transport easily propagates into the lower stratosphere at $4° \times 5°$ resolution and further biases the UTLS $CH_4$ fields. Thus, the weakened tropopause barriers are a major defect of the $4° \times 5°$ model resolution.

Finally, we quantified the impact of the AMF bias on the $XCH_4$ fields. Figure 12 shows change in absolute bias between the model and GOSAT $XCH_4$ in the R1 and R2 experiments. Generally, regridding $2° \times 2.5°$ mass fluxes (first column) had a relatively weak impact on $XCH_4$. Some reduction in the model−GOSAT difference was observed in February at the position of the polar vortex over Europe and North America and in March-May over China. However, this was accompanied by weak increase in the difference in other regions. In the R2 experiment (third column), the reduction in the absolute bias was significantly larger. The transport corrections in R2 reduced the positive $XCH_4$ difference at high latitudes. The mean positive difference over Europe in February was reduced by up to 16 ppb, whereas the positive model−GOSAT difference over China in March-May was reduced by up to 30 ppb. The model bias over China, we argue, was caused by weakened vertical advective transport as a result of a combination of regridding the winds and the strong surface emissions in China that resulted in $CH_4$ being partly trapped in the boundary layer over the continent. Generally, the results suggest that the incorrect hAMFs produce noticeable local biases in the $XCH_4$ fields.

### 5.4.2 Eddy mass flux

The transport correction implemented in the previous section only partly accounted for the missing vertical motion as inferred from Fig. 6 (third row). In this section, we show that the rest of the missing motion can be explained by the loss of tracer eddy mass flux. The continuity equation for the tracer mass in the coarse-resolution model grid box in the absence of sources and sinks is defined as

$$\frac{\partial(q\delta p)}{\partial t} + \boldsymbol{\nabla} \cdot (\overline{q\delta p\mathbf{u}}) = 0, \tag{3}$$

where $q$ is tracer mixing ratio, $\delta p$ is the pressure thickness and $\mathbf{u}$ is the 3D velocity. Assuming that both $q$ and $\delta p\mathbf{u}$ vary on sub-grid scales, Eq. 3 can be rewritten as

$$\frac{\partial(q\delta p)}{\partial t} + \boldsymbol{\nabla} \cdot (\overline{q} \cdot \overline{\delta p\mathbf{u}}) + \boldsymbol{\nabla} \cdot (\overline{q' \cdot (\delta p\mathbf{u})'}) = 0 \tag{4}$$

where $\overline{(\ )}$ is the grid box average and $(\ )'$ is the deviation from the average. The third term on the left-hand side represents the divergence of the tracer eddy mass flux that arises from correlation between $q'$ and $(\delta p\mathbf{u})'$. This divergence term gets lost due to averaging of sub-grid scale fields at coarse resolution.

Vertical transport at middle and high latitudes is largely driven by synoptic scale eddies (extra-tropical cyclones) generated by baroclinic instability (Stohl, 2001; Parazoo et al., 2011). The combined action of convection and advection transports tracers upward and poleward from the surface in warm conveyor belts (WCBs) that originate ahead of cold fronts and flow above warm fronts of cyclones. The WCBs transport air from the boundary layer into the free troposphere, and thereby, are an important mechanism for ventilating the lower troposphere (Kowol-Santen et al., 2001; Sinclair et al., 2008; Ding et al., 2015).





At the same time, these cyclones transport upper tropospheric air downward and equatorward at mid-latitudes, following dry intrusions behind the cold fronts. This creates a mean upward tracer flux for tropospheric species such as $CH_4$ and $^{222}Rn$ that are emitted at the surface. At NH mid-latitudes, regions of extratropical cyclone activity are located in the western part of Atlantic and Pacific oceans and partly overlap with eastern parts of North America and China (Stohl, 2001; Eckhardt et al., 2004; Shaw et al., 2016), which happen to be major $CH_4$ source regions. This makes WCBs particularly important for upward transport of $CH_4$ at mid-latitudes. As shown by Stohl et al. (2002), WCB trajectories over China experience rapid ascent

and end up in the upper troposphere over the western Pacific, whereas WCB trajectories over North America originate in the planetary boundary layer (PBL) and extend to the upper troposphere over Europe.

Frontal uplift, however, may be sensitive to the horizontal resolution of the model. Sinclair et al. (2008) show that the efficiency of the uplift depends strongly on turbulent mixing in the PBL, which raises the tracer to the altitude penetrated by the WCB, and on horizontal Ekman transport, which supplies the tracer into the frontal region. Furthermore, the coarse-

resolution model may not resolve narrow frontal zones and the associated horizontal tracer convergence. Therefore, part of the vertical transport associated with the sub-grid scale correlation between the vAMF anomalies and the tracer mixing ratio anomalies ("anomalies" = deviations from the coarse grid cell average) is lost. Increased numerical diffusion acting on sharp inter-cell concentration gradients in the frontal zone is another issue that may interfere with vertical transport. Horizontal diffusion would create additional horizontal tracer mass flux, which has to be extracted from the vertical mass flux (according

to mass conservation), reducing the altitude of tracer penetration into the free troposphere. In addition to frontal uplift, vertical advective $CH_4$ transport at mid-latitudes is through convergent uplift in the centre of cyclones and orographic uplift on the lee side of mountains. These mechanisms are of less significance, however they may also be sensitive to model resolution.

Figure 13 shows an example of $CH_4$ fields produced by a cyclone passing over the eastern United States modelled at three resolutions. It gives a snapshot of the cyclone at 12:00 UTC on March 21, 2010. The high-resolution ($0.5° \times 0.67°$) case

was simulated using the nested GEOS-Chem model with boundary and initial conditions from the $4° \times 5°$ model run. Both the $4° \times 5°$ and $2° \times 2.5°$ resolution cases have the same initial $CH_4$ conditions on 1 March 2010. Figure 13 shows that at $0.5° \times 0.67°$ resolution, $CH_4$ surface concentrations are higher in the frontal zones and the $CH_4$ plume is lifted higher in the atmosphere along the moist isentropes. Qualitatively, the $2° \times 2.5°$ resolution $CH_4$ fields are more similar to those at the $0.5° \times 0.67°$ resolution than to those at $4° \times 5°$. This may suggest that the $2° \times 2.5°$ resolution model approaches the spatial limit

at which circulation in frontal zones can be resolved, whereas the $4° \times 5°$ resolution is just too coarse for these purposes.

For February-May 2010, we calculated the $CH_4$ and $^{222}Rn$ vertical eddy mass fluxes lost by degrading the model resolution from $2° \times 2.5°$ to $4° \times 5°$. This was done as follows:

1. The $2° \times 2.5°$ vertical tracer and air mass fluxes, and the tracer concentrations were archived from the $2° \times 2.5°$ forward model run and regridded to $4° \times 5°$.

2. The $4° \times 5°$ tracer mass fluxes were defined as a product of the regridded $2° \times 2.5°$ vertical air mass fluxes and the tracer concentrations.





3. The lost eddy mass flux was set equal to the difference between the regridded $2° \times 2.5°$ tracer mass fluxes in Step 1 and those calculated in Step 2.

Figure 14a shows the structure of the $CH_4$ mass flux at $700$ hPa. It can be inferred from the figure that the calculated eddy mass flux is largest in the mid-latitude storm track regions of both hemispheres over eastern South America, South Africa, and particularly, over eastern Asia, however, it is rather weak over North America. It also pronounced over some mountainous regions, especially over the Himalayas, and in the inter-tropical convergence zone (ITCZ) over Africa. Figure 14b shows the monthly $CH_4$ tendency (integral of the eddy mass flux over each global pressure surface) associated with the eddy mass flux, both globally and over North America. For comparison, we also show the $CH_4$ tendency for the eddy mass flux over North America derived from the $0.5° \times 0.67°$ nested $CH_4$ simulation. Generally, because $CH_4$ is well mixed in the troposphere, the tendency terms are rather small. Eddy mass flux acts to reduce $CH_4$ concentrations from approximately $950$ hPa to $600$ hPa by as much as $20$ ppb month$^{-1}$ and increase it in the upper troposphere by up to $7$ ppb month$^{-1}$. It also increases $CH_4$ concentrations near the surface below $950$ hPa. The North American example also shows that the $0.5° \times 0.67°$ $CH_4$ tendency in the lower troposphere is about twice as large as $2° \times 2.5°$ tendency.

We used the calculated eddy mass flux as a correction to the advective tracer mass flux at $4° \times 5°$. Figure 6m-n shows the impact of the combined eddy mass flux correction and the AMF correction (discussed in Sect. 5.4.1) on the zonal structure of the $^{222}$Rn and $CH_4$ fields. The corrected $4° \times 5°$ simulation recovers much of the structure of the $2° \times 2.5°$ fields, so the differences between the "fixed" $4° \times 5°$ simulation and the standard $4° \times 5°$ simulation look similar to the differences between the $2° \times 2.5°$ and $4° \times 5°$ simulations (compare Figs. 6m and 6d). The issue for both the $^{222}$Rn and $CH_4$ simulations at $4° \times 5°$, as noted in Sect. 5.4.1, is that corrections to the vertical transport into the troposphere leaks to the stratosphere, hence the tracer transport at the coarse resolution cannot be fully recovered. Generally, the eddy correction has a smaller impact on the $CH_4$ fields and on the $XCH_4$ distribution (Fig. 12, second column), but has a significant impact on short-lived $^{222}$Rn. We anticipate that the influence of the eddy mass flux would be larger if it were derived from the global native resolution simulation.

## 6 Summary and discussions

We used the GEOS-Chem model at the horizontal resolutions of $4° \times 5°$ and $2° \times 2.5°$ to understand the sources of resolution-induced biases in the model. We focused on the period of February-May 2010 to match the analysis period of Stanevich et al. (submitted), who used a weak constraint 4D-Var assimilation approach to characterize model errors in the GEOS-Chem $CH_4$ simulation. The GEOS-Chem $CH_4$ simulation was evaluated using $XCH_4$ retrievals from TANSO-FTS on-board GOSAT, ground-based $XCH_4$ retrievals from TCCON, and solar occultation $CH_4$ retrievals from ACE-FTS on-board SCISAT. Comparison of the model to all three datasets pointed to the presence of significant transport errors at the $4° \times 5°$ resolution, which were greatly reduced at $2° \times 2.5°$. Discrepancies in the $CH_4$ fields induced by the model resolution included a latitudinal $XCH_4$ bias with large positive $XCH_4$ anomalies at high latitudes and small negative anomalies in the tropics. A significant part of this bias was related to discrepancies in the stratosphere. In addition, a positive $XCH_4$ bias was associated with the





polar vortex. In the troposphere, a positive resolution-induced $XCH_4$ bias in the model−GOSAT differences was also observed over China and shown to be related to reduced vertical transport at $4° \times 5°$. The model evaluation against GOSAT, ACE-FTS and TCCON suggested that a weak latitudinal bias is present in the $2° \times 2.5°$ model and may be related to the stratosphere. We found that the magnitude of the resolution-induced differences between the $4° \times 5°$ and $2° \times 2.5°$ fields was similar in magnitude to the remaining model−GOSAT difference at $2° \times 2.5°$ resolution.

We assessed the impact of the resolution-induced model biases on optimized $CH_4$ surface emissions for February-May 2010
by performing inversion analyses at both model resolutions using the 4D-Var method in GEOS-Chem. The $4° \times 5°$ inversion suggested reduced $CH_4$ emissions at high latitudes and increased emissions in the tropics relative to the $2° \times 2.5°$ model. The differences were large at grid-box scales, but were less than 30% when the inferred emissions were aggregated to the large TransCom regions. The moderate sensitivity of the emissions to the induced biases may be due to limited data density and observational coverage of GOSAT, particularly at high latitudes. However, the sensitivity to model errors is expected to
be higher for data from missions such as TROPOMI, which will provide better observational coverage. Generally, given the magnitude of the model biases, we do not recommend the $4° \times 5°$ GEOS-Chem model for $CH_4$ inverse modelling. Although the estimated model errors are much smaller at the $2° \times 2.5°$ resolution, additional work is needed to better quantify the resolution-induced errors at $2° \times 2.5°$ and assess their potential impact on inferred $CH_4$ source estimates.

Using $^{222}Rn$, $^{7}Be$ and $CH_4$ tracers at the two model resolutions, we investigated the origins of model errors related to
coarsening of the model resolution. The results showed that, fundamentally, the majority of the biases are caused by increased numerical diffusion at the $4° \times 5°$ model resolution. Numerical diffusion acts to smear sharp tracer concentration gradients and is particularly detrimental in regions with strong potential vorticity gradients, including the tropopause layer, boundaries of the tropical pipe, and the polar vortex. The results of this study are consistent with Strahan and Polansky (2006) who also showed that the $4° \times 5°$ resolution model, driven by a similar advection scheme, cannot maintain adequate mixing barriers,
which leads to enhanced stratosphere-troposphere exchange across the tropopause and enhanced mixing in the vicinity of the tropical branch of the Brewer-Dobson circulation and the polar vortex. As a consequence, at $4° \times 5°$, there is less $CH_4$ in the troposphere and more $CH_4$ is mixed into the lower stratosphere at high latitudes. Overall, this produces lower $XCH_4$ fields in the tropics and higher $XCH_4$ at high latitudes.

The tracer experiments also pointed to a weakening of vertical transport at coarser resolution in the troposphere, mainly
at middle to high latitudes. Partly, it was caused by using the coarse resolution wind fields to recalculate the air mass fluxes (AMFs) at the grid box interfaces. Biased AMFs produced non-negligible local biases in the $XCH_4$ fields such as a large positive bias over China. We showed that this problem can be mitigated in GEOS-Chem by archiving and globally remapping the native resolution horizontal AMFs in order to drive advection at the coarse resolution instead of calculating the horizontal AMFs from the coarse-resolution wind fields. The remaining differences in vertical transport were explained by the loss of
tracer eddy mass flux due to coarsening the model resolution and averaging the sub-grid model variability.

GEOS-Chem employs the Lin-Rood scheme for advection and, as mentioned by Prather et al. (2008) and Strahan and Polansky (2006), doubling the resolution of the Lin-Rood scheme from $4° \times 5°$ to $2° \times 2.5°$ may improve the model simulation. However, although doubling the resolution improved the quality of the modelled $CH_4$ fields, numerical biases may still be





present in the $2° \times 2.5°$ resolution. Their importance, particularly, in regards to inverse modeling of $CH_4$ emissions, should be evaluated in future studies using on-line transport in the native general circulation model as a benchmark simulation.

# 7 Data and code availability

The GOSAT satellite data are available at http://www.esa-ghg-cci.org/sites/default/files/documents/public/documents/GHG-CCI_ DATA.html. The TCCON data are available at http://tccondata.org/. The ACE-FTS data are available at https://databace.scisat.
ca/level2/ace_v3.5_v3.6/, and registration is required to download the data. The code for the GEOS-Chem model is publicly available and can be downloaded from www.geos-chem.org. The output from the GEOS-Chem model simulations used in this analysis are available upon request.

*Author contributions.* IS led the study and wrote the paper. DBAJ and KS guided the work and edited the paper. RJP and HB provided GOSAT retrievals. DW, JN, CP, TW, RS, MS, FH, RK, NMD, and VAV provided TCCON data. KAW provided insight into the use of
ACE-FTS data. FD assisted in the initial configuration of the model simulation. All co-authors read and commented on the paper.

*Acknowledgements.* This work was supported by funding from Environment and Climate Change Canada and the Natural Science and Engineering Research Council (NSERC) of Canada. We thank R. J. Parker for providing GOSAT $XCH_4$ data. R. J. Parker was funded via an ESA Living Planet Fellowship with additional funding from the UK National Centre for Earth Observation (NCEO), the ESA Greenhouse Gas Climate Change Initiative (GHG-CCI) and the EU Copernicus Climate Change Service (C3S). We thank the Japanese Aerospace Exploration
Agency, National Institute for Environmental Studies, and the Ministry of Environment for the GOSAT data and their continuous support as part of the Joint Research Agreement. This research used the ALICE High Performance Computing Facility at the University of Leicester for the GOSAT retrievals. Funding for Wollongong TCCON is provided in part by the Australian Research Council (ARC) grants DP160101598, DP140101552, DP110103118 and LE0668470. The Atmospheric Chemistry Experiment (ACE), also known as SCISAT, is a Canadian-led mission mainly supported by the Canadian Space Agency and NSERC.



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





**Table 1.** Evaluation of a priori $4° \times 5°$ and $2° \times 2.5°$ resolution GEOS-Chem model fields against TCCON XCH$_4$ retrievals (mean station-wise statistics for the period of February-May 2010).

| | Mean difference [ppb] | | Standard deviation [ppb] | | Correlation ($R$) | |
|---|---|---|---|---|---|---|
| | $4° \times 5°$ | $2° \times 2.5°$ | $4° \times 5°$ | $2° \times 2.5°$ | $4° \times 5°$ | $2° \times 2.5°$ |
| Sodankyla (67.37°N, 26.63°E) | 30.0 | 15.6 | 18.9 | 11.6 | 0.49 | 0.88 |
| Bialystok (53.23°N, 23.03°E) | 11.9 | 3.7 | 9.3 | 7.7 | 0.39 | 0.67 |
| Bremen (53.10°N, 8.85°E) | 6.3 | 0.0 | 14.3 | 11.0 | -0.37 | 0.34 |
| Karlsruhe (49.10°N, 8.44°E) | 6.4 | -0.2 | 9.6 | 8.7 | 0.32 | 0.52 |
| Orleans (47.97°N, 2.11°E) | 3.9 | 1.7 | 8.9 | 8.4 | 0.31 | 0.46 |
| Garmish (47.48°N, 11.06°E) | 9.9 | -0.1 | 9.0 | 9.7 | 0.46 | 0.53 |
| Park Falls (45.95°N, 90.27°W) | 1.9 | -2.4 | 9.7 | 8.4 | 0.37 | 0.58 |
| Lamont (36.60°N, 97.486°W) | 1.4 | 0.6 | 11.1 | 9.4 | 0.27 | 0.49 |
| Izana (28.30°N, 16.5°W) | -5.8 | -3.2 | 7.6 | 6.8 | 0.64 | 0.68 |
| Wollongong (34.41°S, 150.88°E) | 7.5 | 6.5 | 8.9 | 8.7 | 0.58 | 0.55 |
| Lauder (45.04°S, 169.68°E) | 9.6 | 7.8 | 5.6 | 4.8 | 0.72 | 0.80 |



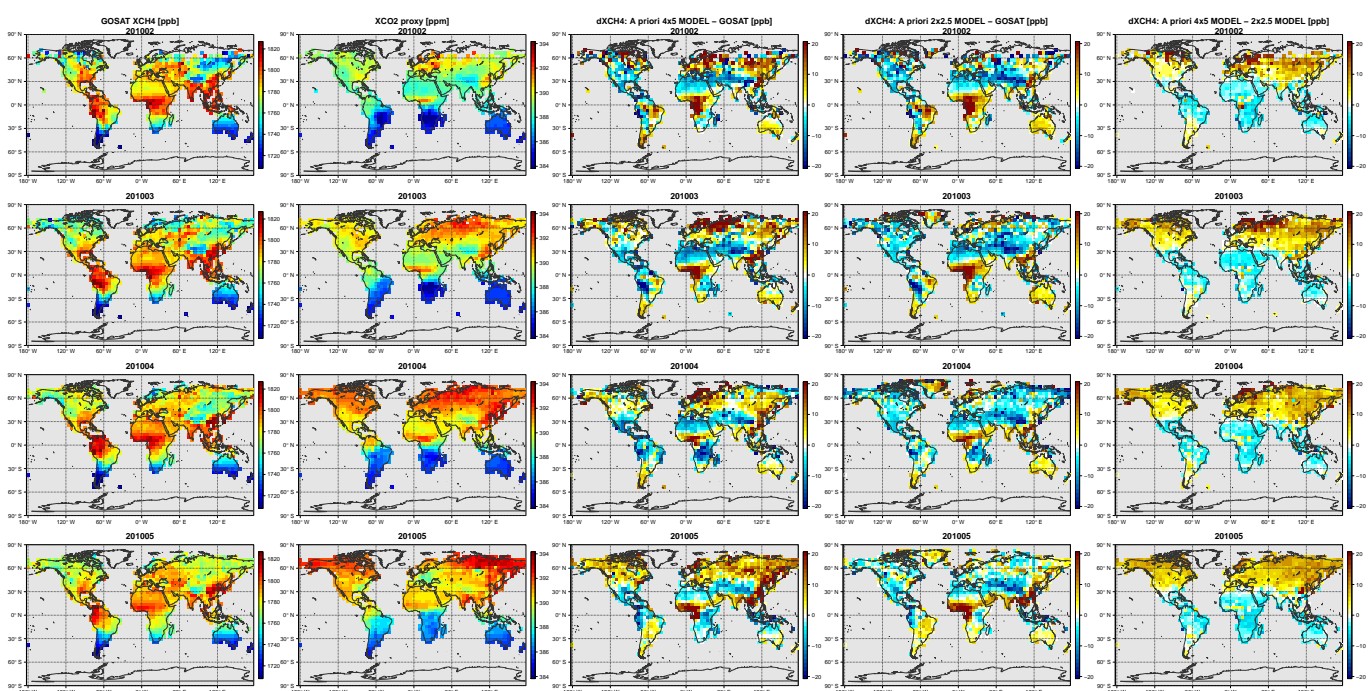

**Figure 1.** Monthly mean fields for February to May 2010 (rows 1 - 4). First column: GOSAT $XCH_4$ retrievals based on the new $XCO_2$ proxy fields. Second column: the new $XCO_2$ proxy fields. Third column: a priori $XCH_4$ difference between the $4° \times 5°$ GEOS-Chem and GOSAT. Fourth column: a priori $XCH_4$ difference between the $2° \times 2.5°$ GEOS-Chem and GOSAT averaged to the $4° \times 5°$ resolution. Fifth column: $XCH_4$ bias calculated as the difference between the GEOS-Chem $4° \times 5°$ and $2° \times 2.5°$ fields co-located with GOSAT observations and smoothed with the GOSAT averaging kernels.



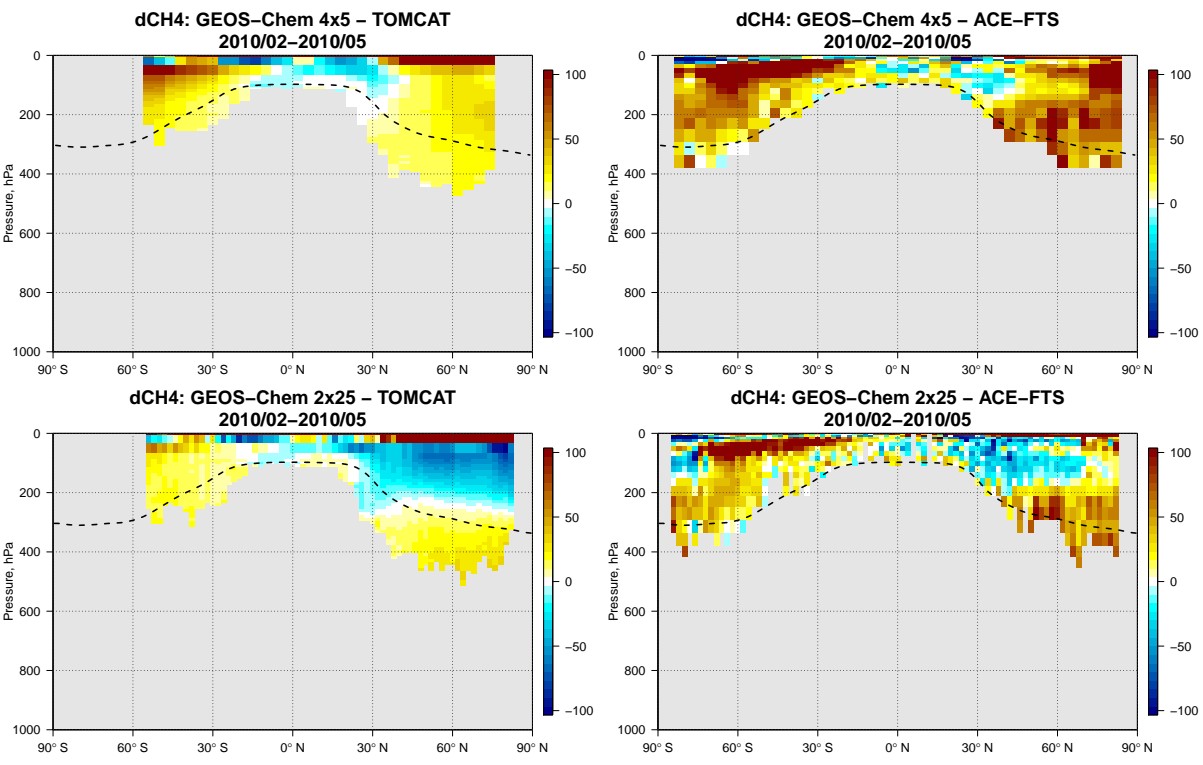

**Figure 2.** Zonal mean $CH_4$ differences (median value for the period of February - May 2010) in the stratosphere between GEOS-Chem at $4°$ $\times 5°$ and the GOSAT $CH_4$ a priori fields (top left), GEOS-Chem at $4° \times 5°$ and ACE-FTS $CH_4$ retrievals (top rght), GEOS-Chem at $2° \times$ $2.5°$ and the GOSAT $CH_4$ a priori fields (bottom left), and between GEOS-Chem at $2° \times 2.5°$ and ACE-FTS $CH_4$ retrievals (bottom right). The dashed line represents the mean dynamic tropopause averaged over February-May 2010 from the archived GEOS-5 meteorological fields.





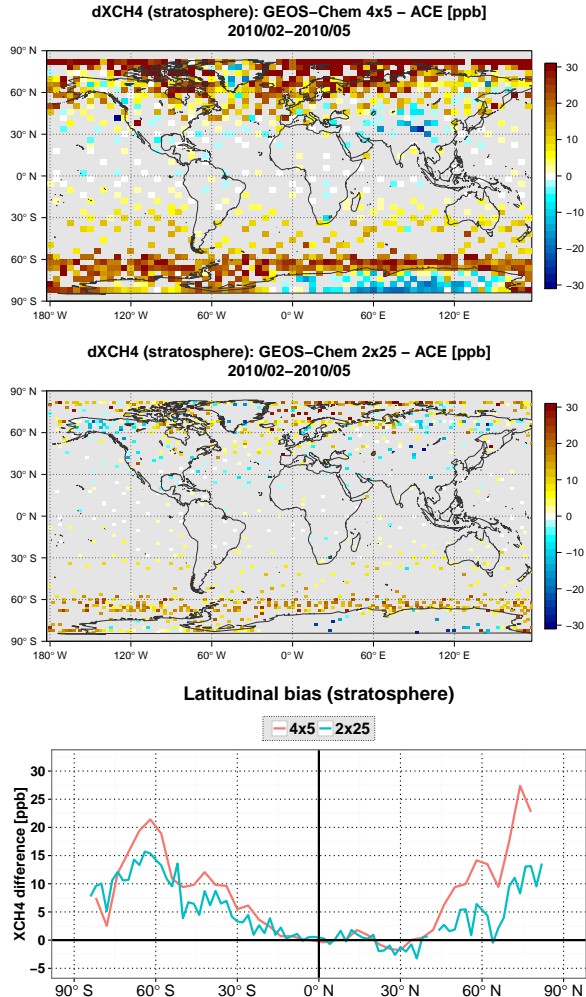

**Figure 3.** XCH$_4$ difference between GEOS-Chem and ACE-FTS due to the stratosphere, for GEOS-Chem at $4° \times 5°$ (top panel) and at $2° \times 2.5°$ (middle panel). The zonal mean difference for both model resolutions are shown in the bottom panel. XCH$_4$ differences were obtained by augmenting the ACE-FTS stratospheric profile with the GEOS-Chem troposphere and smoothing the vertical CH$_4$ difference profile with the mean zonal GOSAT averaging kernels.





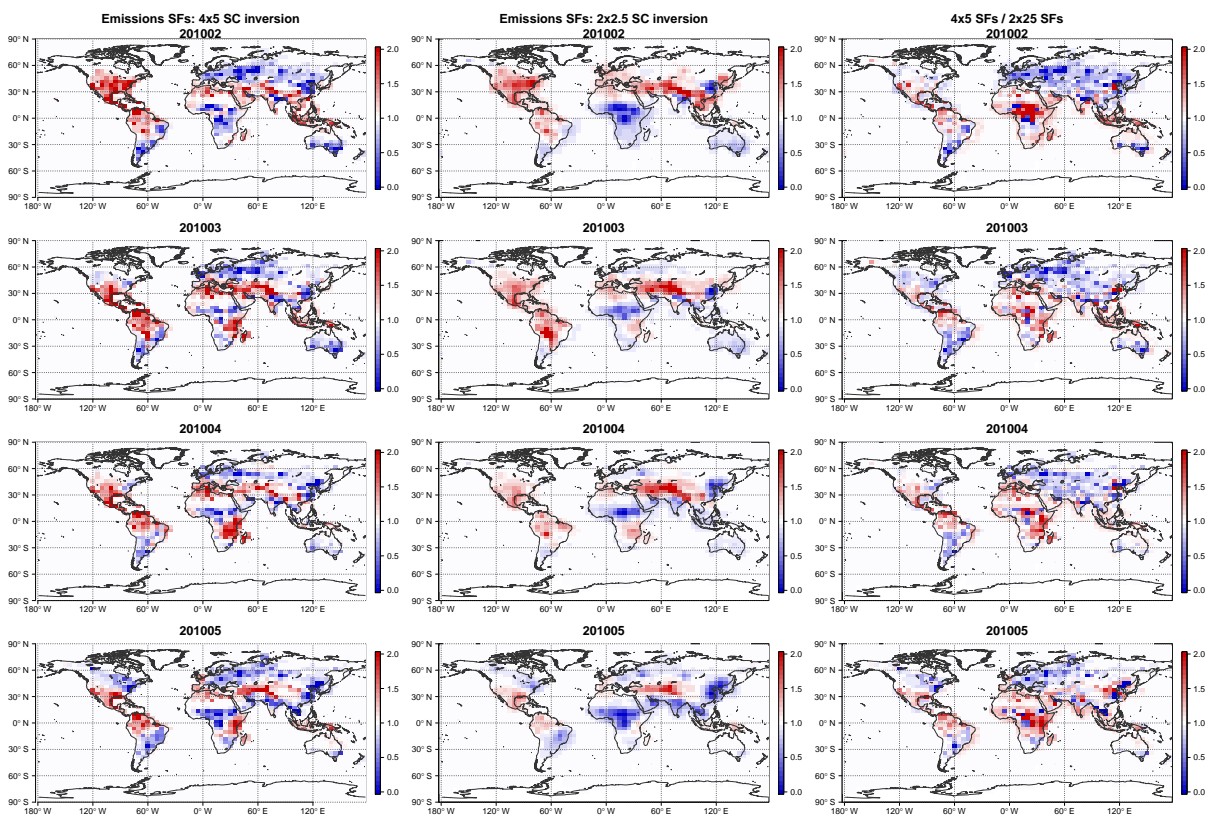

**Figure 4.** Monthly mean scaling factors (ratio of optimized to a priori CH$_4$ surface emissions). Shown are the scaling factors from the $4°\times5°$ (first column) and $2°\times2.5°$ (second column) assimilation. The third column shows the ratio of the $4°\times5°$ to $2°\times2.5°$ scaling factors. Results are shown for February (first row), March (second row), April (third row), and May (fourth row) 2010.





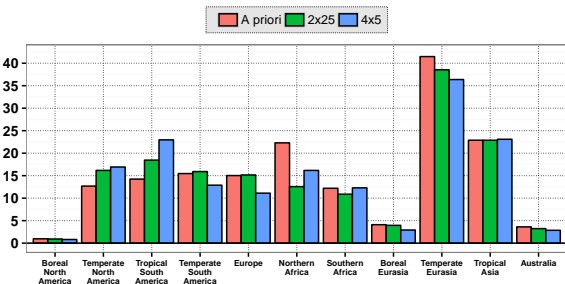

**Figure 5.** Total CH$_4$ emissions in the 11 TransCom land regions for the period of February-May 2010. Shown are the a priori emissions (red) and optimized emissions using the $4° × 5°$ ( blue) and $2° × 2.5°$ (green) versions of the GEOS-Chem model.



**Figure 6.** Evaluation of the zonal mean distribution of $^{222}$Rn (left column), $^{7}$Be (middle column), and CH$_4$ (right column) in March 2010 in GEOS-Chem. First row: the zonal mean tracer concentrations. Second row: the mean difference between the 2° × 2.5° and the 4° × 5° simulations. Third row: the mean difference between the R1 experiment and the 4° × 5° simulations. Fourth row: the mean difference between the R2 experiment and the 4° × 5° simulations. Fifth row: the mean difference between the R3 experiment and the 4° × 5° simulations. R1 is the 4° × 5° simulation driven with regirdded 2° × 2.5° hAMFs, R2 is the 4° × 5° simulation with the hAMFs over NA, EU and CH regions replaced by regridded 0.5° × 0.67° hAMFs, and R3 is the 4° × 5° simulation driven with regirdded 2° × 2.5° hAMFs and with an additional CH$_4$ eddy mass flux based on the 2° × 2.5° simulation.





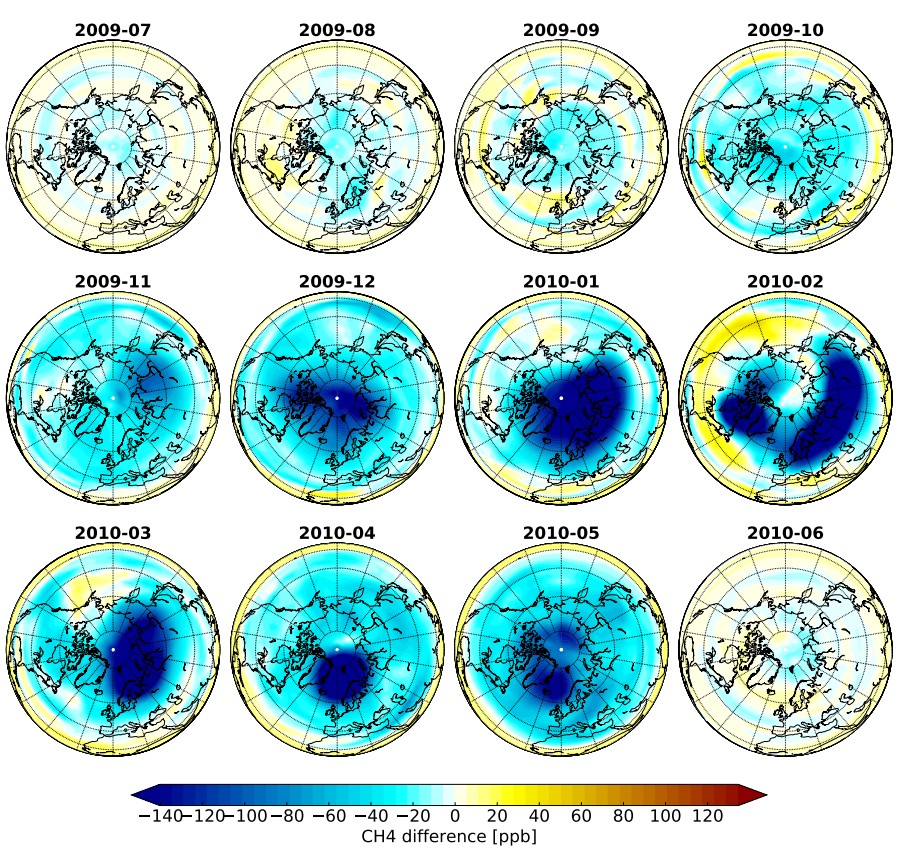

**Figure 7.** Monthly mean difference in CH$_4$ (in ppb) at 50 hPa (North Pole projection) between the GEOS-Chem $2° \times 2.5°$ and $4° \times 5°$ simulations.



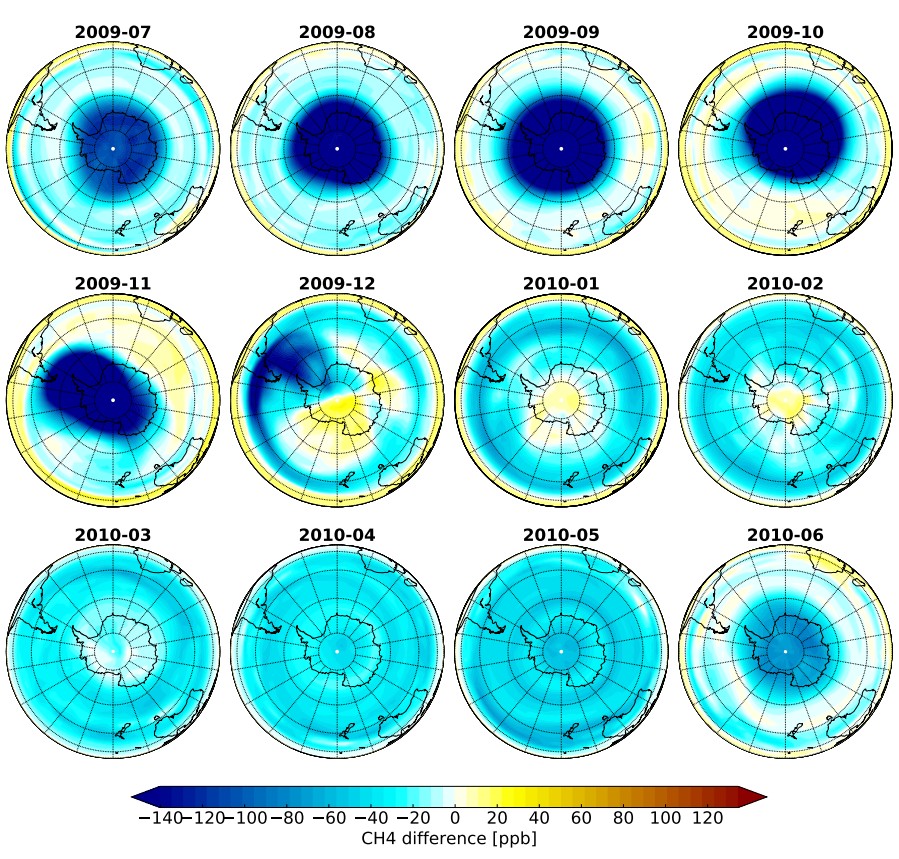

**Figure 8.** Monthly mean difference in CH$_4$ (in ppb) at 50 hPa (South Pole projection) between the GEOS-Chem $2° \times 2.5°$ and $4° \times 5°$ simulations.





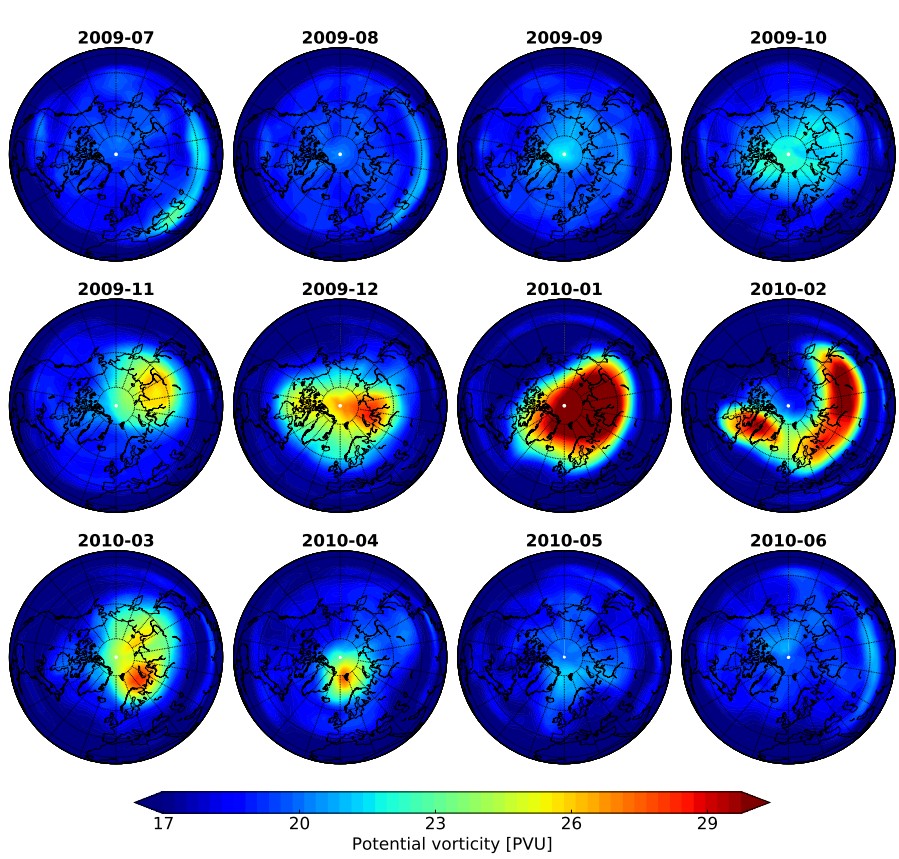

**Figure 9.** Mean monthly potential vorticity on the 450 K isentropic surface from the archived GEOS-5 meteorological fields (north pole projection).



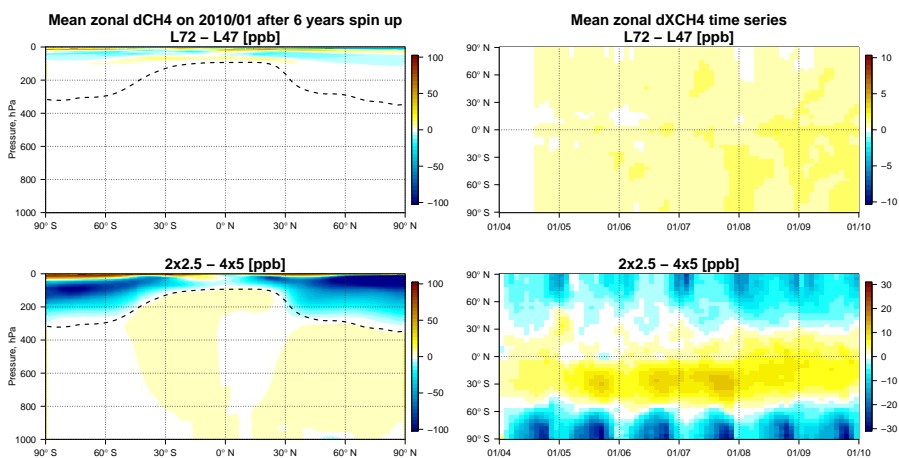

**Figure 10.** First row: differences in CH$_4$ between the GEOS-Chem simulations with 72 and 47 vertical levels. Second row: differences in CH$_4$ between the GEOS-Chem simualtions at $2° \times 2.5°$ and $4° \times 5°$. Left column: the zonal mean CH$_4$ difference after 6 years of simulation. Right column: the zonal mean XCH$_4$ bias time series obtained by smoothing the CH$_4$ difference profiles with mean latitudinal GOSAT averaging kernels.





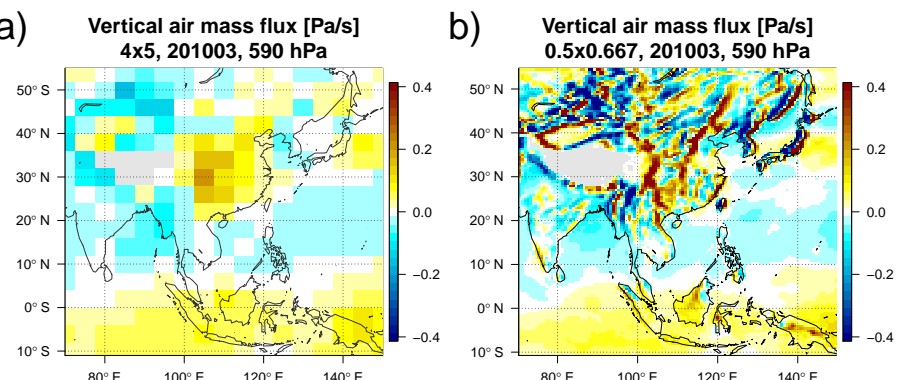

**Figure 11.** Mean advective vAMFs at $4° \times 5°$ and $0.5° \times 0.67°$ resolution in March 2010 at 590 hPa.



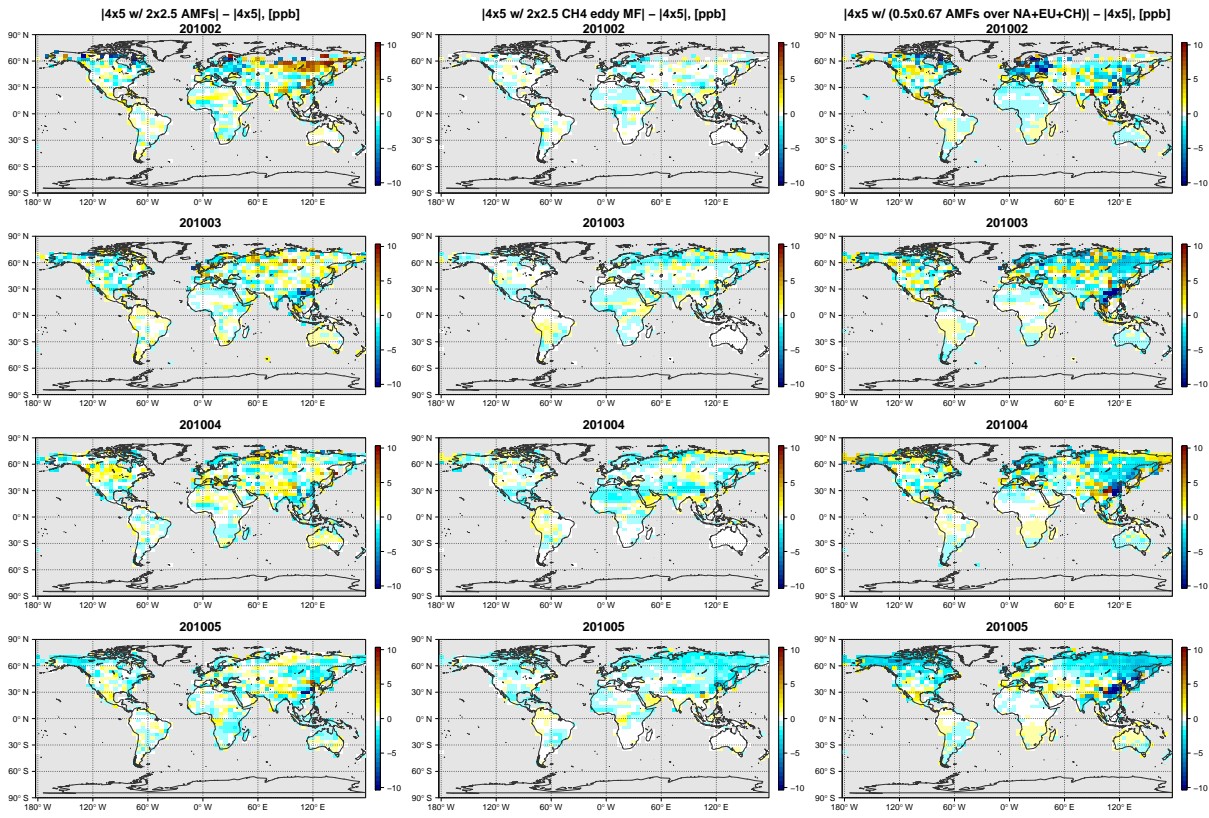

**Figure 12.** Change in the monthly mean absolute bias between the model and GOSAT XCH$_4$ in the R1 and R2 experiments for February to May 2010 (rows 1 - 4). Left column: changes in the absolute bias for the R1 experiment (the "fixed" $4° \times 5°$ simulation), which is driven by the regridded $2° \times 2.5°$ hAMFs. Middle column: changes in the absolute bias for the R1 experiment with the additional CH$_4$ eddy mass flux based on $2° \times 2.5°$ simulation. Right column:changes in the absolute bias for the R2 experiment, which is the "fixed" simulation with the hAMFs over North America, Europe, and Asia replaced by the regridded $0.5° \times 0.67°$ hAMFs. See the text in Section 5.4.1 for more details about the R1 and R2 experiments.



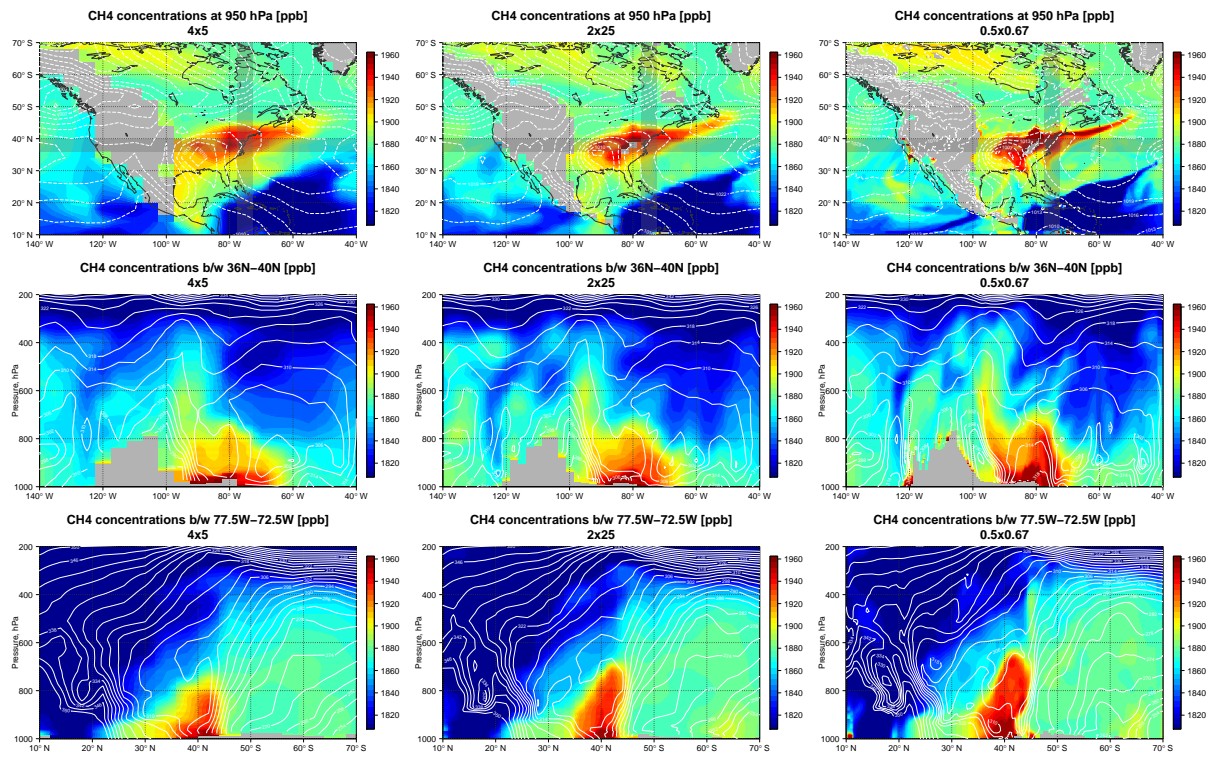

**Figure 13.** The distribution of CH$_4$ in GEOS-Chem at 12:00 UTC on 21 March 2010 at $4° \times 5°$ (left column), $2° \times 2.5°$ (middle column), and $0.5° \times 0.67°$ (right column). Top row: the distribution at 950 hPa, with sea level pressure indicated by the long-dashed white lines. Middle row: altitude-longitude cross section along the grey horizontal band in the figure in the top row. Bottom row: altitude-latitude cross section along the grey vertical band in the figure in the top row. The solid white lines in the figures in the middle and bottom rows indicate the moist isentropes.



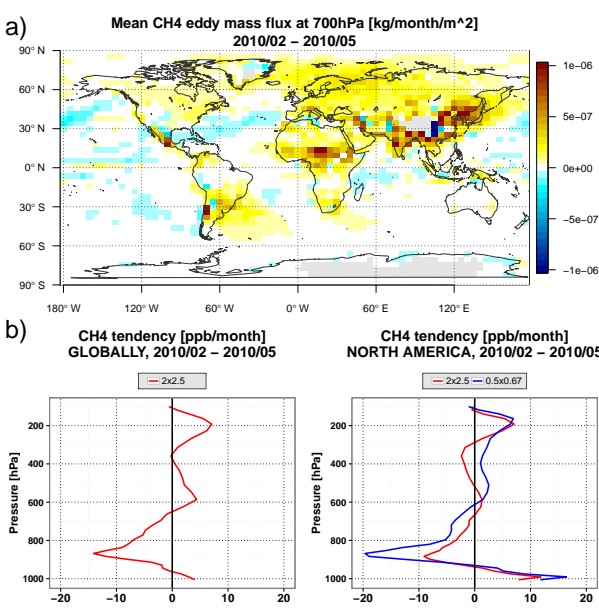

**Figure 14.** (a) The mean CH$_4$ eddy mass flux at 700 hPa for February-May 2010 lost by degrading the model resolution from $2° \times 2.5°$ to $4° \times 5°$. (b) Left panel: the vertical profile of the globally averaged CH$_4$ tendency (in ppb/month) caused by the CH$_4$ eddy mass flux lost by degrading the model from $2° \times 2.5°$ to $4° \times 5°$. Right panel: the vertical profile of the CH$_4$ tendency over North America caused by the eddy mass flux lost by degrading the model resolution from $2° \times 2.5°$ to $4° \times 5°$ (red) and from $0.5° \times 0.67°$ to $4° \times 5°$ (blue).