# Peer review of "Characterizing model errors in chemical transport modeling of methane: Impact of model resolution in versions v9-02 of GEOS-Chem and v35j of its adjoint model"

_Geoscientific Model Development, 2019_

## Referee Comment (RC1) · Anonymous Referee #1 · 6 Jan 2020

Overview: The manuscript "Characterizing model errors in chemical transport modeling of methane: Impact of model resolution in versions v9-02 of GEOS-Chem and v35j of its adjoint model" by Stenevich et al. describes the effect of model resolution on the forward and inverse modelling of methane in the GEOS-Chem model, and the potential effect on the model's estimation of posterior fluxes after data assimilation. The forward model was used to run short simulations of methane, which were then compared to a number of remote sensing sources. Following this, simulations of species representative of atmospheric transport at various scales were carried out in order to assess the

causes of the differences in transport caused by the changing resolution. The paper suggests that, for this particular model at least, the resolution of the simulations is very important and could significantly affect conclusions drawn through forward and inverse modelling using GEOS-Chem.

Overall the manuscript is very well written, with few technical corrections necessary. The figures are generally quite clear and well chosen, although some further detail needs to be provided for some of them. The methods used within the manuscript are appropriate for such a study, are able to provide assessment of the effect of model resolution on inverse modelling, and should provide pause for thought for all modellers. Although the paper is relatively niche in its focus, which describes the effects of transport parameterisations which are specific to GEOS-Chem, it does flag up an issue which will be of interest to the GEOS-Chem modelling community, and might cause other modellers to assess these issues within their own models.

My main reservation is that the quantity of plots is quite large, whilst the quality of some of them is a little low. The authors have a tendency to overwhelm the reader with information in these plots, at the risk of the main message being lost. Generally, the clarity of figures should be maximised, both in terms of literal clarity and clarity of message. Once this is fixed, I suggest that this paper is suitable for publication in this journal as long as the following small revisions are carried out.

Comments:

Figure 1: There is a lot of information here, so each individual plot is quite small. This means that the panel labels and the labels of the colour bars are illegible. I'd suggest making the text for each panel larger. I'd also suggest that, for improved clarity, the CO2 panels could be removed from this figure.

Figures 7 - 9: These plots could easily be combined by reducing the number of panels in each plot. Figures 7 and 9, for example, do not require the NH summer panels to be included, whilst Figure 9 also shows a number of panels with no features.

Figure 12: similar to figure 1, most figure text is unclear. Also, I'd suggest that clearer context here would be provided by showing the differences between the 'fixed' model simulations and the satellite, rather than the 'fixed' and 'unfixed' model simulations. As it stands, it is difficult to see what improvements are being provided by the fixes.

Figure 13: whilst the grey bars are visible on my computer screen, they were not visible at all when I printed out the manuscript, so I'd suggest that a clearer method should be used to indicate the regions displayed in the second and third rows here.

Technical corrections: Page 1 line 8: columns abundances -> column abundances

Figure 2 caption: rght -> right

Page 8 line 19: missing reference

Page 8 line 23: relative -> relative to

Figure 3 bottom panel label (and other figures): 25 -> 2.5

Figure 10 caption: note different colour bar scales for panels 2 and 4
* * *

---

## Referee Comment (RC2) · Anonymous Referee #2 · 14 Jan 2020

General Comments

The manuscript 'Characterizing model errors in chemical transport modelling of methane: Impact of model resolution in version v9-02 of GEOS-Chem and v35j of its adjoint model' by Stanevich et al. describes the impact of model resolution on tracer transport and optimised surface CH4 fluxes in GEOS-Chem. Initially, model simulations of CH4 at two different resolutions were compared with satellite and TCCON data. Large differences in simulated methane column abundances were found between the two resolutions, which resulted in differences in optimised regional methane fluxes of

up to 30%. Further simulations using additional tracers then assessed the causes of differences in transport between the two resolutions. Results from this study suggest that model resolution can have a significant impact on the location and magnitude of optimised methane fluxes, introducing a latitudinal bias.

The manuscript is in general well-structured and well-written. The figures are useful and mostly clear, although the text could do with enlarging on many of them. Although the manuscript is model-specific, the results are relevant and significant for the methane modelling community, highlighting the potential magnitude and importance of model errors when trying to constrain the methane budget. I recommend this manuscript for publication following the minor revisions outlined below.

Specific / Technical Comments

Page 1, line 8: change 'columns' to 'column'

Page 1, line 15: The sentence starting 'We also identified..' repeats what is said in the previous sentence.

Page 3, line 33: change 'winds field' to wind fields'

Page 4, line 6: change 'Because' to 'As' or alternative.

Page 5, line 33: How is soil absorption determined / or can a reference be provided for this?

Figure 1: The labels are currently too small to read. The XCO2 proxy fields could be removed to allow better viewing of the remaining plots.

Page 7, Section 3.1: As I understand it, the authors spun up the model using standard GEOS-Chem methane emissions for 5.5 years, then switched to optimised emissions (from the 4 x 5 degree resolution model) for the remaining 7 months of the initialisation period, and then for the analysis period switched back to using the standard GEOS-Chem emissions. If the magnitude of the differences between the optimised and standard emissions is similar to those shown in Figure 4, the changes are quite large, and seven months is short relative to the methane lifetime; it would be helpful here if the authors could clarify the aim and influence of switching to optimised emissions for 7 months before the analysis period?

Page 8, line 19: reference missing (' A similar stratospheric bias was reported by ??').

Page 11, line 22: change to 'be a useful indicator'

Page 13, lines 5-10, Section 5.3: How were these simulations initialised and what period were they run for? Do they use the optimised emissions? Quite a few different simulations are described in this paper, perhaps a table listing all/a subset of the different simulations and their set ups would help.

Figure 13: I cannot see a grey vertical band or read the labels on the white contours. Fewer contours with larger labels could help make them more readable.

Page 17, line 12: Mention 'R3' in the text here (simulation R3 is used in the label and caption in Fig 6, but not specifically mentioned in the text).
* * *

---

## Author Response (AR1)

We thank the two referees for their helpful comments. In particular, we are appreciative of their suggestions for improving the figures. Below, we respond to each of their comments, with the original comments in italics.

**Referee #1**

*Overview: The manuscript "Characterizing model errors in chemical transport modeling of methane: Impact of model resolution in versions v9-02 of GEOS-Chem and v35j of its adjoint model" by Stenevich et al. describes the effect of model resolution on the forward and inverse modelling of methane in the GEOS-Chem model, and the potential effect on the model's estimation of posterior fluxes after data assimilation. The forward model was used to run short simulations of methane, which were then compared to a number of remote sensing sources. Following this, simulations of species representative of atmospheric transport at various scales were carried out in order to assess the causes of the differences in transport caused by the changing resolution. The paper suggests that, for this particular model at least, the resolution of the simulations is very important and could significantly affect conclusions drawn through forward and inverse modelling using GEOS-Chem.*

*Overall the manuscript is very well written, with few technical corrections necessary. The figures are generally quite clear and well chosen, although some further detail needs to be provided for some of them. The methods used within the manuscript are appropriate for such a study, are able to provide assessment of the effect of model resolution on inverse modelling, and should provide pause for thought for all modellers. Although the paper is relatively niche in its focus, which describes the effects of trans- port parameterisations which are specific to GEOS-Chem, it does flag up an issue which will be of interest to the GEOS-Chem modelling community, and might cause other modellers to assess these issues within their own models.*

*My main reservation is that the quantity of plots is quite large, whilst the quality of some of them is a little low. The authors have a tendency to overwhelm the reader with information in these plots, at the risk of the main message being lost. Generally, the clarity of figures should be maximised, both in terms of literal clarity and clarity of message. Once this is fixed, I suggest that this paper is suitable for publication in this journal as long as the following small revisions are carried out.*

*Comments:*

*Figure 1: There is a lot of information here, so each individual plot is quite small. This means that the panel labels and the labels of the colour bars are illegible. I'd suggest making the text for each panel larger. I'd also suggest that, for improved clarity, the CO2 panels could be removed from this figure.*

We have revised the figure with larger fonts. We have also removed the CO2 panels as well as the panels for April 2010. In revising the figure, we changed the layout so that the columns now correspond to the months for which results are shown.

*Figures 7 - 9: These plots could easily be combined by reducing the number of panels in each plot. Figures 7 and 9, for example, do not require the NH summer panels to be included, whilst Figure 9 also shows a number of panels with no features.*

We have removed the summer plots (July – October). We have also removed Figure 8 (the SH plots) from the manuscript.

*Figure 12: similar to figure 1, most figure text is unclear. Also, I'd suggest that clearer context here would be provided by showing the differences between the 'fixed' model simulations and the satellite, rather than the 'fixed' and 'unfixed' model simulations. As it stands, it is difficult to see what improvements are being provided by the fixes.*

To enhance clarity, we have increased the size of the fonts. We have also removed the April 2010 data to reduce the plot density and to match the time period presented in Figure 1. As regards the comparison with the satellite, the plot is showing the change in the bias between the model and the satellite. It is difficult to see the improvement because the impact of the R1 experiment (shown in columns 1 and 2) on the model bias was "relatively weak", as we noted on page 15, line 6, of the original manuscript. The R2 experiment (column 3) produced "significantly larger" bias reductions.

*Figure 13: whilst the grey bars are visible on my computer screen, they were not visible at all when I printed out the manuscript, so I'd suggest that a clearer method should be used to indicate the regions displayed in the second and third rows here.*

We have increased the contour spacing and font size. We also enhanced the grey bands to make them sharper and more visible.

*Technical corrections:*

*Page 1 line 8: columns abundances -> column abundances*

The text was changed.

*Figure 2 caption: rght -> right*

Corrected.

*Page 8 line 19: missing reference*

The reference was added.

*Page 8 line 23: relative -> relative to*

Corrected.

*Figure 3 bottom panel label (and other figures): 25 -> 2.5*

Changed.

*Figure 10 caption: note different colour bar scales for panels 2 and 4*

Because of the differences in magnitude between the two sources of bias, it is not helpful to plot them on the same scale. If we use the colour scale from panel 4 for the plot in panel 2, much of the plot will be white. As a result, we decided not to change the figure.

**Referee #2**

*General Comments*

*The manuscript 'Characterizing model errors in chemical transport modelling of methane: Impact of model resolution in version v9-02 of GEOS-Chem and v35j of its adjoint model' by Stanevich et al. describes the impact of model resolution on tracer transport and optimised surface CH4 fluxes in GEOS-Chem. Initially, model simulations of CH4 at two different resolutions were compared with satellite and TCCON data. Large differences in simulated methane column abundances were found between the two resolutions, which resulted in differences in optimised regional methane fluxes of up to 30%. Further simulations using additional tracers then assessed the causes of differences in transport between the two resolutions. Results from this study suggest that model resolution can have a significant impact on the location and magnitude of optimised methane fluxes, introducing a latitudinal bias.*

*The manuscript is in general well-structured and well-written. The figures are useful and mostly clear, although the text could do with enlarging on many of them. Although the manuscript is model-specific, the results are relevant and significant for the methane modelling community, highlighting the potential magnitude and importance of model errors when trying to constrain the methane budget. I recommend this manuscript for publication following the minor revisions outlined below.*

*Specific/Technical Comments*

*Page 1, line 8: change 'columns' to 'column'*

We have changed it.

*Page 1, line 15: The sentence starting 'We also identified..' repeats what is said in the previous sentence.*

Thank you. We have deleted the sentence.

*Page 3, line 33: change 'winds field' to wind fields'*

Corrected.

*Page 4, line 6: change 'Because' to 'As' or alternative.*

It was unclear what was wrong with this sentence. Nevertheless, we changed "Because" to "Since".

*Page 5, line 33: How is soil absorption determined / or can a reference be provided for this?*

We have added a reference.

*Figure 1: The labels are currently too small to read. The XCO2 proxy fields could be removed to allow better viewing of the remaining plots.*

The figure has been revised. See our response to Referee #1.

*Page 7, Section 3.1: As I understand it, the authors spun up the model using standard GEOS-Chem methane emissions for 5.5 years, then switched to optimised emissions (from the 4 x 5*

*degree resolution model) for the remaining 7 months of the initialisation period, and then for the analysis period switched back to using the standard GEOS- Chem emissions. If the magnitude of the differences between the optimised and standard emissions is similar to those shown in Figure 4, the changes are quite large, and seven months is short relative to the methane lifetime; it would be helpful here if the authors could clarify the aim and influence of switching to optimised emissions for 7 months before the analysis period?*

We chose to use optimize emissions for the last seven months of the spin-up period to obtain initial conditions that were in closer agreement with the GOSAT data. We have added a sentence to the text (page 7, line 15) stating this. Seven months is a short period compared to the methane lifetime, but the relevant timescales are the transport timescales not the methane lifetime. Seven months is not long enough to correct for discrepancies in the interhemispheric gradient in the model or for errors in the stratosphere. However, given the timescale for hemispheric mixing, seven months is sufficiently long to obtain an improved description of methane over the major source regions, which was our objective. This was necessary since the goal of the analysis was to understand the source of the model biases associated with transport errors and not errors in the emissions.

The changes in the emissions in Figure 4 are indeed large, particularly for the 4x5 inversion, but they do not accurately reflect the errors in the a priori emissions. As we emphasized in this section, by increasing the resolution to 2x2.5 we significantly reduced the magnitude of the changes in the emissions. In our companion paper (Stanevich et al., Characterizing model errors in chemical transport modelling of methane: Using GOSAT XCH4 data with weak constraint four-dimensional variational data assimilation, submitted to ACP) we showed that the emission estimates are overadjusted to compensate for transport biases. We have therefore added the following text to page 10, line 27: "As discussed in Stanevich et al. (submitted), these large changes in the emissions may reflect an overadjustment of the emissions to compensate for transport-related biases in the model."

*Page 8, line 19: reference missing ('A similar stratospheric bias was reported by ??').*

We have added the reference.

*Page 11, line 22: change to 'be a useful indicator'*

Corrected.

*Page 13, lines 5-10, Section 5.3: How were these simulations initialised and what period were they run for? Do they use the optimised emissions? Quite a few different simulations are described in this paper, perhaps a table listing all/a subset of the different simulations and their set ups would help.*

We started with an initial condition from the free-running model, without any assimilation. The initial condition with 47 vertical levels was regridded in a mass conserving way to 72 levels and also to a horizontal resolution of 2x2.5. This ensured that all simulations began with identical initial conditions. We have added the following text on page 13 explaining this: "The initial condition was selected from the free-running version of the model with 47 vertical levels and regridded to 72 levels and also to a horizontal resolution of 2x2.5. We ensured that mass was conserved so that all simulations began with identical initial conditions." Since the manuscript is already long, we decided against adding an additional table.

*Figure 13: I cannot see a grey vertical band or read the labels on the white contours. Fewer contours with larger labels could help make them more readable.*

As we noted in our response to Referee #1, the figure was revised with fewer contours and enhanced gray bands to make them sharper.

*Page 17, line 12: Mention 'R3' in the text here (simulation R3 is used in the label and caption in Fig 6, but not specifically mentioned in the text).*

We now mention the R3 experiment in the beginning of the last paragraph in Section 5.4.2.

[revised manuscript text omitted]
 $4° \times 5°$ simulations. R1 is the $4° \times 5°$ simulation driven with regirdded $2° \times 2.5°$ hAMFs, R2 is the $4° \times 5°$ simulation with the hAMFs over NA, EU and CH regions replaced by regridded $0.5° \times 0.67°$ hAMFs, and R3 is the $4° \times 5°$ simulation driven with regirdded $2° \times 2.5°$ hAMFs and with an additional CH$_4$ eddy mass flux based on the $2° \times 2.5°$ simulation.

[Figure]

**Figure 7.** Monthly mean difference in CH$_4$ (in ppb) at 50 hPa (North Pole projection) between the GEOS-Chem 2° × 2.5° and 4° × 5° simulations.

Monthly mean difference in CH₄ (in ppb) at 50 (South Pole projection) between the GEOS-Chem 2× 2.5and 4× 5simulations.

[Figure]

**Figure 8.** Mean monthly potential vorticity on the 450 K isentropic surface from the archived GEOS-5 meteorological fields (north pole projection).

[Figure]

**Figure 9.** First row: differences in CH$_4$ between the GEOS-Chem simulations with 72 and 47 vertical levels. Second row: differences in CH$_4$ between the GEOS-Chem simualtions at $2° \times 2.5°$ and $4° \times 5°$. Left column: the zonal mean CH$_4$ difference after 6 years of simulation. Right column: the zonal mean XCH$_4$ bias time series obtained by smoothing the CH$_4$ difference profiles with mean latitudinal GOSAT averaging kernels.

[Figure]

**Figure 10.** Mean advective vAMFs at $4° \times 5°$ and $0.5° \times 0.67°$ resolution in March 2010 at 590 hPa.

[Figure]

**Figure 11.** Change in the monthly mean absolute bias between the model and GOSAT XCH$_4$ in the R1 and R2 experiments for February  March, and May 2010 (rows 1 - 3). Left column: changes in the absolute bias for the R1 experiment (the "fixed" $4° \times 5°$ simulation), which is driven by the regridded $2° \times 2.5°$ hAMFs. Middle column: changes in the absolute bias for the R1 experiment with the additional CH$_4$ eddy mass flux based on $2° \times 2.5°$ simulation. Right column:changes in the absolute bias for the R2 experiment, which is the "fixed" simulation with the hAMFs over North America, Europe, and Asia replaced by the regridded $0.5° \times 0.67°$ hAMFs. See the text in Section 5.4.1 for more details about the R1 and R2 experiments.

[Figure]

**Figure 12.** The distribution of CH$_4$ in GEOS-Chem at 12:00 UTC on 21 March 2010 at $4° × 5°$ (left column), $2° × 2.5°$ (middle column), and $0.5° × 0.67°$ (right column). Top row: the distribution at 950 hPa, with sea level pressure indicated by the long-dashed white lines. Middle row: altitude-longitude cross section along the grey horizontal band in the figure in the top row. Bottom row: altitude-latitude cross section along the grey vertical band in the figure in the top row. The solid white lines in the figures in the middle and bottom rows indicate the moist isentropes.

[Figure]

**Figure 13.** (a) The mean CH$_4$ eddy mass flux at 700 hPa for February-May 2010 lost by degrading the model resolution from $2° \times 2.5°$ to $4° \times 5°$. (b) Left panel: the vertical profile of the globally averaged CH$_4$ tendency (in ppb/month) caused by the CH$_4$ eddy mass flux lost by degrading the model from $2° \times 2.5°$ to $4° \times 5°$. Right panel: the vertical profile of the CH$_4$ tendency over North America caused by the eddy mass flux lost by degrading the model resolution from $2° \times 2.5°$ to $4° \times 5°$ (red) and from $0.5° \times 0.67°$ to $4° \times 5°$ (blue).